# Impact of boundary conditions on the modeled thermal regime of the Antarctic ice sheet

In-Woo Park[1], Emilia Kyung Jin[2], Mathieu Morlighem[3], and Kang-Kun Lee[1]

[1]School of Earth and Environmental Sciences, Seoul National University, Seoul, South Korea
[2]Korea Polar Research Institute, Incheon, South Korea
[3]Department of Earth Sciences, Dartmouth College, Hanover NH, USA

**Correspondence:** jin@kopri.re.kr

**Abstract.** A realistic initialization of ice flow models is critical for predicting future changes in ice sheet mass balance and their associated contribution to sea level rise. The initial thermal state of an ice sheet is particularly important as it controls ice viscosity and basal conditions, thereby influencing the overall ice velocity. Englacial and subglacial conditions, however, remain poorly understood due to insufficient direct measurements, which complicates the initialization and validation of thermal
models. Here, we investigate the impact of using different geothermal heat flux (GHF) datasets and vertical velocity profiles on the thermal state of the Antarctic ice sheet, and compare our modeled temperatures to in situ measurements from 15 boreholes. We find that the temperature profile is more sensitive to vertical velocity than GHF. The basal temperature of grounded ice and the amount of basal melting are influenced by both selection of GHF and vertical velocity. More importantly, we find that the standard approach, which consists of combining basal sliding speed and incompressibility to derive vertical velocities,
provides reasonably good results in fast flow regions (ice velocity $> 50$ m yr$^{-1}$), but performs poorly in in slow flow regions (ice velocity $< 50$ m yr$^{-1}$). Furthermore, the modeled temperature profiles in ice streams, where bed geometry is generated using mass conservation approach, show better agreement with observed borehole temperatures, compared to kriging-based bed geometry.

## 1   Introduction

Global warming has been responsible for rapid sea level rise from the mass loss of ice sheets and glaciers over the past few decades. The mass loss of the Antarctic ice sheet has more than tripled over the past three decades (IPCC AR6 Chapter 9; Fox-Kemper et al., 2021). The retrograde bed slopes in deep submarine basins (Schoof, 2007), the intrusion of warm water in ice shelf cavities (Alley et al., 2016), and the collapse of ice shelves can accelerate this mass loss (Scambos, 2004), especially in West Antarctica. Ice sheet models have been developed to capture these processes (e.g., Larour et al., 2012b; Gillet-Chaulet
et al., 2012; Pollard and DeConto, 2012) and provide projections of future contributions of the ice sheets to sea level rise under different warming scenarios (DeConto and Pollard, 2016; Seroussi et al., 2020). However, the uncertainty in these projections remains high partly due to poorly constrained model inputs, such as bed geometry, basal conditions, ice mechanical properties, or oversimplified parameterization of melting rates under floating ice shelves (e.g., Schlegel et al., 2013; Brondex et al., 2019).

A critical aspect of ice sheet models is their initial conditions. Several important properties, such as ice elevation and surface ice velocity, can be directly observed at the surface of the ice sheet, whereas observing englacial and subglacial properties, such as ice temperature and geothermal heat flux, remain particularly challenging, and direct measurements of these properties are scarce.

In order to get reasonable estimates of these englacial and subglacial fields, inversion techniques are routinely employed to estimate basal friction and ice shelf rigidity (MacAyeal, 1993; Khazendar et al., 2007; Morlighem et al., 2010; Gillet-Chaulet, 2020). These inverse modeling approaches have not been applied to the ice thermal regime of the ice sheet, which remains highly uncertain despite its critical control on ice viscosity and basal friction. Critically, the geothermal heat flux (GHF) is an important parameter that affects basal temperature, water production, and ice dynamics (Pattyn et al., 2008; Seroussi et al., 2017; Smith-Johnsen et al., 2020b); yet, large uncertainties in spatial variation and magnitude of GHFs in Antarctica still remain.

Previous studies have attempted to infer the GHFs using different methods such as a seismic model (Shapiro and Ritzwoller, 2004; An et al., 2015), magnetic satellite data (Maule et al., 2005), and a combination of seismic and magnetic satellite data (Martos et al., 2017). The most accurate measurements are from in situ borehole measurements of temperature profiles that can be used to constrain the GHFs (Dahl-Jensen et al., 1999; Mony et al., 2020; Talalay et al., 2020).

While drilling boreholes requires a lot of resources and efforts, the boreholes provide critical insights into subsurface conditions and lead to a better understanding of the current subglacial and englacial environments as well as past climate (Augustin and Antonelli, 2002; Motoyama, 2007; Slawny et al., 2014; Fisher et al., 2015; Priscu et al., 2021; Mulvaney et al., 2021; Smith et al., 2021). Borehole temperature profiles can also be utilized to validate thermo-mechanical ice sheet models. As boreholes provide vertical temperature profiles, a one-dimensional thermal model is generally utilized to estimate GHFs (Mony et al., 2020) and reconstruct past climates (Zagorodnov et al., 2012; Yang et al., 2018). Since one-dimensional thermal models typically neglect horizontal advection and only consider vertical advection and diffusion (Engelhardt, 2004a; Mony et al., 2020; Talalay et al., 2020), one-dimensional thermal models have strong limitations and may not be applicable in regions of fast flow. The vertical velocities used in one-dimensional thermal model are generally recovered through the equation of incompressibility, assuming a stationary bed and no sliding (Hindmarsh, 1999). Only a handful of three-dimensional thermo-mechanical ice sheet models have utilized these borehole temperature profiles for validation (Joughin et al., 2004; Pattyn, 2010; Seroussi et al., 2013). Moreover, measurements of borehole temperatures in fast flow regions remain scarce due to technical difficulty of drilling boreholes in these regions (Engelhardt, 2004b; Doyle et al., 2018; Anker et al., 2021).

In addition to being sensitive to the GHF, the ice thermal regime is also particularly sensitive to horizontal and vertical ice velocities. While surface ice velocities can be spatially and temporally observed through satellite remote sensing (Mouginot et al., 2012; Derkacheva et al., 2020), englacial velocities are difficult to observe remotely. Few measurements of internal vertical ice velocities are available through direct methods, such as optic-fiber instruments (Pettit et al., 2011), and borehole optical televiewer (OPTV) logging (Hubbard et al., 2020), or indirect methods, such as phase-sensitive radio echo sounder (Gillet-Chaulet et al., 2011; Kingslake et al., 2014). Due to scarcities of internal ice velocity measurements, three-dimensional mechanical models, such as Higher-Order (HO; Pattyn, 2003) and Full Stokes (FS), are used to estimate internal ice velocities

(Pattyn, 2003; Larour et al., 2012b). The ice velocities from mechanical models can, in turn, be used as input variables to
compute three-dimensional ice temperature.

Overall, the difficulty in estimating GHF combined with the lack of observations of subsurface ice velocities and temperature limits our ability to capture the thermal regime of the ice sheet and increases the uncertainty in future mass projections. Here, we perform a suite of sensitivity experiments using a three-dimensional thermo-mechanical model using various GHF sources and different approaches to construct vertical ice velocities. We then compare each modeled temperature to 15 temperature profiles
from in situ borehole drilling campaigns, including 3 boreholes located in fast flow regions to determine which combination of parameters best reproduces measured temperature profiles.

## 2   Methods

### 2.1   Ice flow model

We use the Ice-sheet and Sea-level System Model (ISSM) to model the stress balance and thermal state across the entire
Antarctic continent (Larour et al., 2012b). We rely on an anisotropic mesh with a resolution varying from 2 km in coastal regions to 40 km near ice divides, and refine the mesh to 2 km mesh around the locations of boreholes where temperature measurements are available. The mesh comprises a total of over a million prismatic elements distributed vertically over 15 layers. We use a three-dimensional Higher-Order model (HO; Pattyn, 2003) and assume that the ice viscosity follows Glen's flow (Glen, 1955):

$$\mu = \frac{B}{2\,\dot{\varepsilon}_e^{\frac{n-1}{n}}} \tag{1}$$


where $B$ is the ice rigidity (Pa s$^{-1/3}$), $\dot{\varepsilon}_e$ is the effective strain rate (s$^{-1}$), and $n$ is Glen's law exponent, whose value is 3 in this study. We also utilize the Budd type friction law (Budd et al., 1979; Morlighem et al., 2010):

$$\boldsymbol{\tau}_b = -\alpha^2 N\,\boldsymbol{v}_b \tag{2}$$

where $\alpha$ is the friction coefficient (yr$^{0.5}$ m$^{-0.5}$), $N$ is the effective pressure (taken here as simply $\rho_i g H + \rho_w g \max(0, b)$), and
$\boldsymbol{v}_b$ is the basal ice velocity vector. $\rho_i$ is the ice density, $\rho_w$ is the water density, $H$ is the ice thickness, and $b$ is the bed elevation with respect to sea level. The friction coefficient under grounded ice and the ice rigidity of floating ice shelves are estimated based on an inverse method (Morlighem et al., 2010). To minimize misfit between modeled and observed ice velocities, the surface ice velocity of MEaSUREs version 2 (Rignot, 2017) is used. The ice rigidity under grounded ice is estimated using the temperature-rigidity relation (Cuffey and Paterson, 2010, pp. 72–77).
We use an enthalpy model that considers the transition between cold and temperate ice as well as the conservation of the total energy balance (Aschwanden et al., 2012; Seroussi et al., 2013; Kleiner et al., 2015). Here, the enthalpy model is referred to as the thermal model and assumes that the ice is in thermal steady-state:

$$0 = -\boldsymbol{v} \cdot \nabla E + \phi_i + \begin{cases} \nabla \cdot \left( \dfrac{k_i}{c_i \rho_i} \nabla E \right), & \text{if } E < E_s \\[2ex] \nabla \cdot (k \nabla T_{pmp} + k_0 \nabla E), & \text{if } E \geq E_s \end{cases} \tag{3}$$

where $\boldsymbol{v} = (v_x, v_y, v_z)$ is the ice velocity vector, $E$ is the enthalpy, $\phi_i$ is the internal deformation heat, $E_s$ is the enthalpy of pure ice, $k = (1 - \omega)k_i + \omega k_w$ is the mixture thermal conductivity (with $\omega$ representing water content), $k_i$ and $k_w$ are the thermal conductivity of pure ice and liquid water, $k_0$ is a small positive constant (Aschwanden et al., 2012), $c_i$ is the heat capacity of ice, and $T_{pmp}$ is the pressure melting point of ice.

The surface temperature is constrained using mean 2-m air temperature data from ERA-Interim, which assimilated the recent atmospheric conditions from 1979 to 2018 with a $0.125° \times 0.125°$ resolution (Dee et al., 2011). At the bottom, we impose a Neumann boundary condition with a heat flux from GHF and frictional heating. The basal temperature under floating ice shelves is set to the pressure melting point. An anisotropic Streamline Upwind Petrov–Galerkin (SUPG) method is adopted since it is more accurate than the original SUPG scheme, which is sensitive to low aspect ratios between the horizontal and vertical resolution meshes (Rückamp et al., 2020). The stress balance and thermal state are closely coupled because the internal deformation and frictional heat from the stress balance affect the thermal model. In turn, the ice rigidity inferred from the thermal model influences the stress balance model. To capture this coupling and reach thermo-mechanical consistency, we iterate 10 times by solving iteratively the stress balance and thermal model until it reaches convergence. The convergence is reached when the difference in mean basal temperature was lower than $0.5°C$ between two consecutive iterations.

We use the surface elevation from the Reference Elevation Model of Antarctica (REMA; Howat et al., 2019). The bed geometry is from BedMachine version 1 (Morlighem et al., 2020), which used the mass conservation method to generate the bed geometry in fast flow regions and streamline diffusion slow flow regions (Morlighem et al., 2010).

## 2.2 Vertical velocities

We compute the thermal state of the ice sheet using two different vertical velocity profiles: 1) vertical velocity computed by solving for incompressibility while accounting for the inferred basal sliding (hereafter IVz), and 2) the equation of incompressibility of ice while not allowing basal sliding when surface ice velocities are below 10 m yr$^{-1}$ (hereafter IVz-nosliding). In other words, IVz-nosliding ignores the inferred basal sliding velocities from the initial inversion and assumes that the bed is frozen when surface velocities are $< 10$ m yr$^{-1}$.

For IVz and IVz-nosliding, we recovered the vertical velocity from the continuity equation as follows:

$$v_z(z) = v_z(b) + \int_b^z -\frac{\partial v_x}{\partial x} - \frac{\partial v_y}{\partial y} \, dz' \tag{4}$$

For IVz-nosliding, we set $v_{x,y}(b) = 0$, while for IVz, the basal vertical velocity is set as:

$$v_z(b) = v_x(b) \frac{\partial b}{\partial x} + v_y(b) \frac{\partial b}{\partial y} - \dot{M}_b \tag{5}$$

where $\dot{M}_b$ is the basal melting rate (in m yr$^{-1}$ ice equivalent).

## 2.3 Geothermal heat flux

We compare four different geothermal flux datasets: Shapiro and Ritzwoller (2004) (SR), which used a seismic model to extrapolate heat-flow measurements, 2) Maule et al. (2005) (Maule), which used a magnetic model with satellite magnetic data, 3) An et al. (2015) (An), which used a crust-lithosphere temperature model, and 4) Martos et al. (2017) (Martos), which inferred the GHF by compiling aeromagnetic data. The mean GHF on grounded ice is 60.78 mW m$^{-2}$ for SR, 65.61 mW m$^{-2}$ for Maule, 54.66 mW m$^{-2}$ for An, and 65.49 mW m$^{-2}$ for Martos.

## 2.4 Borehole temperature measurements

To validate the thermal models, we compile all available borehole temperature profiles from Dome Fuji (Hondoh et al., 2002), Styx Glacier (Yang et al., 2018), the WAIS Divide (Cuffey and Clow, 2014), the Bruce Plateau (Zagorodnov et al., 2012), Law Dome (Van Ommen et al., 1999; Dahl-Jensen et al., 1999), and the WAIS discharge to the Ross ice shelf (Engelhardt, 2004b) (Table 1). The boreholes in the West Antarctica Ice Sheet region are drilled at Whillans Ice Stream (WIS), Bindschadler Ice Stream (BIS), Engelhardt Ridge (ER), Kamb Ice Stream (KIS), Raymond Ridge (RR), Unicorn (UC), Alley Ice Stream (AIS), and Siple Dome (SD) (Engelhardt, 2004a) (Figure. 1b). We use here borehole names from Engelhardt (2004b): ER-1996-12, SD-1997-1, RR-1997-42, KIS-1996-2, KIS-2000-1,2, UC-1993-11, UC-1993-14, AIS/WIS-1991-1, AIS/WIS-1995-4,7, and BIS-1998-4,5.

Since the vertical distance between temperature measurements along the borehole profile and triangular mesh are not uniform, we calculate a weighted absolute misfit between the modeled and measured temperatures (or modeled ice surface velocities) when evaluating the thermal model's performance:

$$\text{misfit} = \sum_{i=1}^{n_{obs}} w_i \left| Y_i^{mod} - Y_i^{obs} \right| \tag{6}$$

where $n_{obs}$ is the number of measured points at each borehole (or the number of observed ice velocities), $i$ indicates the index of the specific measured elevation (or index of the ice velocity area), $w_i$ is a weight calculated from the ratio of a specific measured point's occupying length to the total measured length (or ratio of the measured area to the total area), and $Y_i$ is the temperature (or ice velocity magnitude). The subscripts $obs$ and $mod$ indicate the observed and modeled variables, respectively.

Since the ice thickness and the surface temperature of the ice flow model are not always exactly consistent with the observed borehole data, we make adjustments using an exponential decaying correction following Pattyn (2010):

$$X_{corr} = X + (X_0 - X) \exp\left( -\frac{\sqrt{(x - x_w)^2 + (y - y_w)^2}}{\sigma} \right) \tag{7}$$

where $(x_w, y_w)$ is the location of the borehole, $X_0$ is the observed quantity, and $X$ is the model ice thickness or surface temperature. The surface temperature at each borehole, except for SD-1997-1, RR-1997-42, UC-1993-11, AIS/WIS-1988-

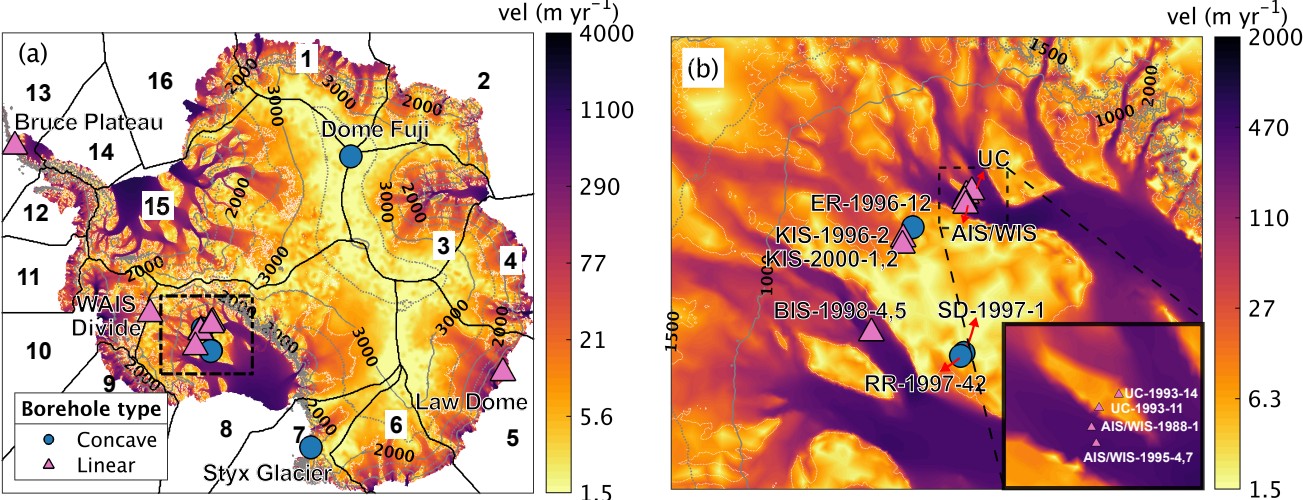

**Figure 1.** (a) Borehole locations with temperature measurements overlaid over ice velocity (Rignot, 2017). The black dashed box shows the location of (b). The black solid box in (a) indicates each basin from Jourdain et al. (2020), and each number indicates each basin number. We use different symbols for each borehole based on the shape of their temperature profile (triangle and cross red dots indicate concave and linear profiles, respectively). The gray contours indicate surface elevations, with dash lines for every 500 m and solid lines for every 1000 m. The white dot contours indicate regions where ice velocity is 10 m yr$^{-1}$. (b) Enlargement of borehole locations at West Antarctica overlain over the ice velocity. The borehole names are abbreviated: WIS, Whillans Ice Stream; BIS, Bindschadler Ice Stream; ER, Engelhardt Ridge; KIS, Kamb Ice Stream; RR, Raymond Ridge; UC, Unicorn; AIS, Alley Ice Stream; SD, Siple Dome.

1, and BIS-1998-4,5, are corrected where given climatological temperature is relatively higher than the observed surface temperature (Table 1). $X_{corr}$ is the corrected data, and $\sigma$ is the radius of influence, which is here set to 50 km. The geometry from BedMachine is constrained using radar-derived ice thickness measurements, except for that at Dome Fuji, and Law Dome, for which the mapping remained largely unconstrained. These two locations are the only places where an ice thickness correction is applied so that the ice thickness is 3,090 m at Dome Fuji, and 1,220 m at Law Dome, respectively.

## 3 Results

### 3.1 Model experiments

To estimate the ice temperature of the entire Antarctic continent, we perform eight different experiments by combining two different vertical velocity profiles (IVz and IVz-nosliding) and four different GHF datasets. Table 2 shows the weighted absolute misfits between the modeled and observed surface ice velocities across the entire domain. The mean ice surface velocity misfit is 12.45 m yr$^{-1}$ for the IVz group, and 19.53 m yr$^{-1}$ for the IVz-nosliding group. The standard deviation of the ice velocity misfit for the IVz group, 0.09 m yr$^{-1}$, is relatively lower than that of the IVz-nosliding group, 0.35 m yr$^{-1}$.

| Name | Latitude | Longitude | Surface temperature (°C) | Ice velocity (m yr$^{-1}$) | Drilled depth (m) | Ice thickness (m) | Date | Reference |
|---|---|---|---|---|---|---|---|---|
| | | | | | | | | |
| Slow flow region | | | | | | | | |
| Dome Fuji | 77º19'1"S | 39º42'12"E | -57.3 | 0.3(3.3) | 3035.2 | 3028±15[a] | 1996 Dec | Hondoh et al. (2002) |
| SD-1997-1 | 81º39'30"S | 211º11'30"E | -24.55 | 0.5(3.0) | | 1004.6 | 1997 Nov | Engelhardt (2004b) |
| RR-1997-42 | 81º35'47"S | 211º18'22"E | -24.55 | 2.0(4.0) | | 955.0 | 1998 Jan | Engelhardt (2004b) |
| Styx Glacier | 73º51'6"S | 163º41'13.20"E | -31.8 | 3.7(5.1) | 210.5[b] | 550[c] | 2016 Nov | Yang et al. (2018) |
| UC-1993-11 | 83º34'56"S | 221º51'15"E | -25.09 | 6.4(8.9) | | 910.6 | 1993 Dec | Engelhardt (2004b) |
| UC-1993-14 | 83º40'45"S | 221º37'42"E | -25.09 | 7.4(5.5) | | 1091.6 | 1994 Jan | Engelhardt (2004b) |
| WAIS Divide | 79º28'0"S | 112º4'60"W | -29.97 | 12.1(15.3) | 3405[d] | 3455[e] | 2006-2011 | Slawny et al. (2014) |
| Law Dome | 66º46'11"S | 112º48'25"E | -21.8 | 8.3(12.2) | 1195.6[f] | 1220±25[g] | 1996-1997 | Van Ommen et al. (1999) |
| Bruce Plateau | 66º1'12"S | 295º57'36"E | -14.8 | 49.13(25.9) | 447.65[h] | 447[h] | 2010 Feb | Zagorodnov et al. (2012) |
| ER-1996-12 | 82º40'36"S | 224º10'29"E | -25.85 | 9.2(6.8) | | 1123.9 | 1997 Jan | Engelhardt (2004b) |
| KIS-1996-2 | 82º26'42"S | 224º1'24"E | -26.92 | 8.9(5.4) | | 1189.0 | 1996 Nov | Engelhardt (2004b) |
| KIS-2000-1,2 | 82º22'0"S | 223º35'60"E | -25.5 | 2.5(4.7) | | 949.4 | 2000 Dec | Engelhardt (2004b) |
| Siple coast fast flow region | | | | | | | | |
| AIS/WIS-1988-1 | 83º29'58"S | 221º34'34"E | -25.52 | 365(6.1) | | 1035.0 | 1988 Dec | Engelhardt (2004b) |
| AIS/WIS-1995-4,7 | 83º27'43"S | 221º3'13"E | -24.94 | 379(7.2) | | 1026.3 | 1997 Jan | Engelhardt (2004b) |
| BIS-1998-4,5 | 81º4'25"S | 219º59'41"E | -24.35 | 376(3.9) | | 1086.0 | 1999 Jan | Engelhardt (2004b) |

**Table 1.** Summary of each borehole information. Observed ice velocity is from Rignot (2017), and parenthesis indicates error in magnitude of ice velocity. The date refers to when the boreholes were drilled. [a]Parrenin et al. (2007); [b]Yang et al. (2018); [c]Hur (2013); [d]Slawny et al. (2014); [e]WAIS Divide Project Members (2013); [f]Morgan et al. (1997); [g]Zagorodnov et al. (2012).

| GHF | Vertical velocity | |
|---|---|---|
| | IVz | IVz-nosliding |
| SR | SR-IVz (12.43 m yr$^{-1}$) | SR-IVz-nosliding (19.88 m yr$^{-1}$) |
| Maule | Maule-IVz (12.46 m yr$^{-1}$) | Maule-IVz-nosliding (19.05 m yr$^{-1}$) |
| An | An-IVz (12.56 m yr$^{-1}$) | An-IVz-nosliding (18.53 m yr$^{-1}$) |
| Martos | Martos-IVz (12.34 m yr$^{-1}$) | Martos-IVz-nosliding (19.65 m yr$^{-1}$) |

**Table 2.** Experimental design for eight simulations using different vertical velocities and geothermal heat fluxes. The value between parentheses under each experiment represents the weighted absolute misfit between observed and modeled surface ice velocity across the entire domain.

Figure 2 displays the measured and modeled vertical profiles of the ice temperature at the 15 borehole locations (see Figure S1). The measured vertical profiles of the borehole temperatures, marked as black dashed lines in Figure 2, can be categorized into two groups based on temperature profile shapes. One group exhibits concave profiles, for which the vertical advection toward the bed dominates, while the other group has more linear shape, for which vertical diffusion dominates. Dome Fuji, SD-1997-1, RR-1997-42, ER-1996-12, and Styx Glacier at slow flow regions show diffusion dominant temperature profiles compared to the WAIS Divide, Bruce Plateau, Law Dome, KIS-1996-2, KIS-2000-1,2, UC-1993-11, and UC-1993-14, where the advection toward the bed dominates. Note that AIS/WIS-1991-1, AIS/WIS-1995-4,7, and BIS-1998-4,5 are located in regions with comparatively high ice velocity compared to other boreholes and have concave temperature profiles. To clearly define this specific fast flow region, we refer to AIS/WIS and BIS as Siple coast fast flow region.

## 3.2  Borehole temperature profiles

To provide a quantitative comparison between the modeled and observed borehole temperatures, a weighted absolute misfit is calculated (Figure 3). The average temperature misfit values for IVz and IVz-nosliding are 5.64°C and 3.61°C, respectively, and 1.69°C and 2.50°C for slow and Siple coast fast flow regions. The temperature misfit value of the IVz-nosliding group is lower than that of the IVz group, however, the misfit temperatures in the Siple coast fast flow regions for IVz and IVz-nosliding are not exactly the same. The spread in misfits among the different vertical velocity schemes is larger than the one obtained when varying GHF. This shows that the difference in GHF has a limited influence on estimating the overall temperature profiles, while the choice of vertical velocities has a stronger impact. Both the IVz and IVz-nosliding groups demonstrate good performance in Siple coast fast flow regions, such as AIS/WIS and BIS. In the case of slow flow regions, the thermal model's performance for the IVz-nosliding group is improved compared to the IVz group, and the model produced a reduced temperature misfit. A more detailed description of misfit values for each borehole can be found in the next section.

Let's first focus on the first three borehole profiles: SD, RR, and Dome Fuji. They all have linear temperature profiles, which are rarely observed in general borehole temperature profiles. SD and RR are adjacent to each other, but measurements of borehole temperatures at RR are limited to the top few hundred meters. Dome Fuji is located in the interior of the ice sheet. For these boreholes, IVz-nosliding group does not capture the linear shape of the temperature profiles. The IVz-nosliding group at these boreholes has a misfit value within 2°C, which of value is lower than that of IVz-nosliding (Figure 3). The basal temperatures from the IVz-nosliding group reach the pressure melting point at SD, RR (Engelhardt, 2004b), and Dome Fuji. In the case of An, the GHF at each borehole is 40.1 mW m$^{-2}$ for Dome Fuji, 64.9 mW m$^{-2}$ for SD, and 65.3 mW m$^{-2}$ for RR, which are lower than the values from other GHF sources. The basal modeled temperature at An is the lowest, and does not reach pressure melting point. The depth-averaged vertical velocity at Dome Fuji is -0.14 m yr$^{-1}$ for IVz (where a negative value means the vector is oriented downward), which is a higher value than that of IVz-nosliding (-0.01 m yr$^{-1}$) (Table 3). The depth-averaged vertical velocities of IVz at SD and RR are also higher than that of IVz-nosliding. This suggests a larger advection toward the ice sheet base in the IVz group, where downward heat advection is more dominant than the diffusion process, and leads to a colder basal temperature compared to the ones in the IVz-nosliding group.

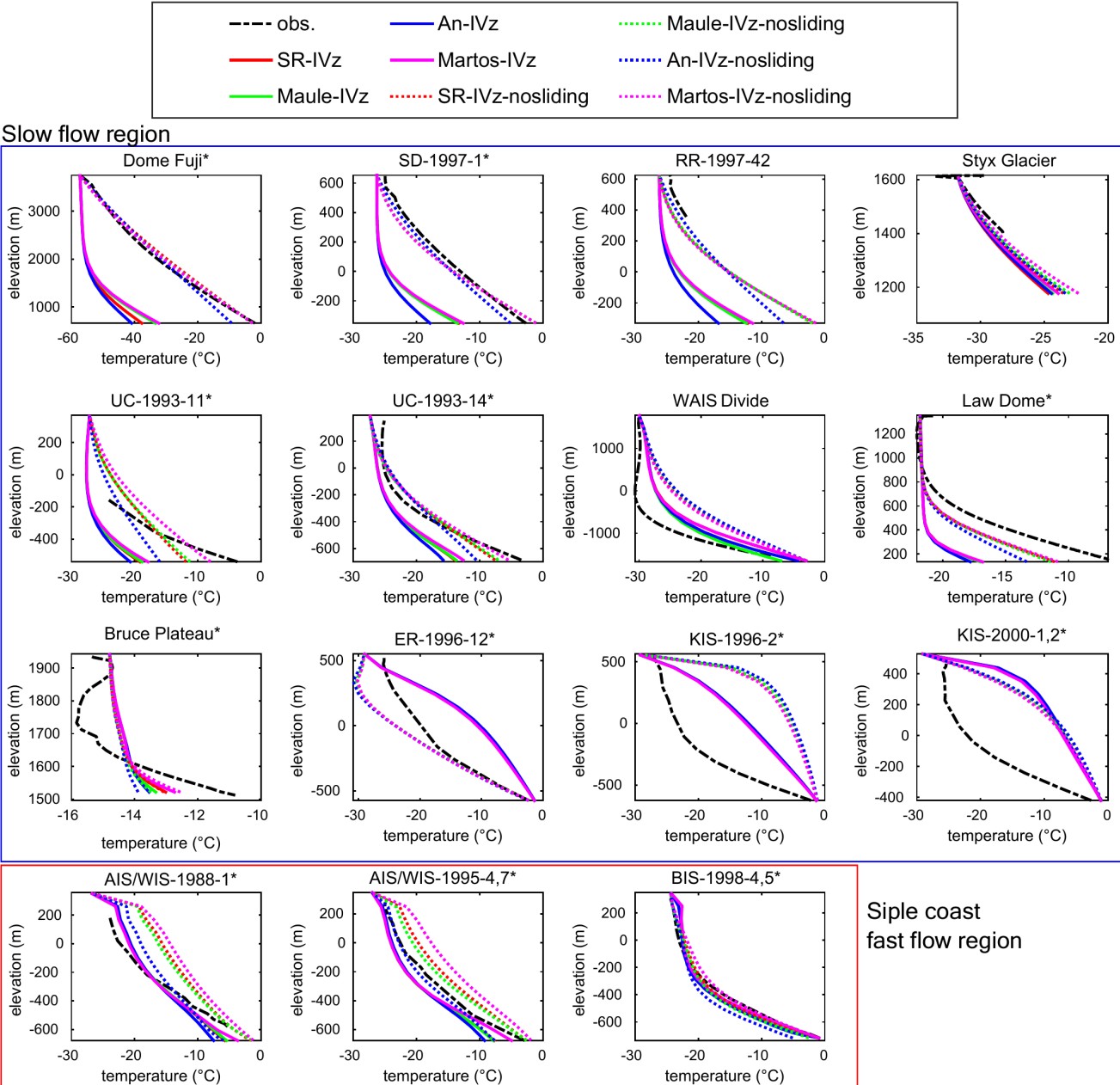

**Figure 2.** Observed and modeled vertical temperature profiles from eight different experiments at 15 borehole locations. Blue and red boxes indicate slow flow and Siple coast fast flow regions, respectively. The bottom elevation at each borehole is set with considering the ice thickness, as listed in Table 1. An asterisk on borehole name indicates that the drilling reaches the bed rock.

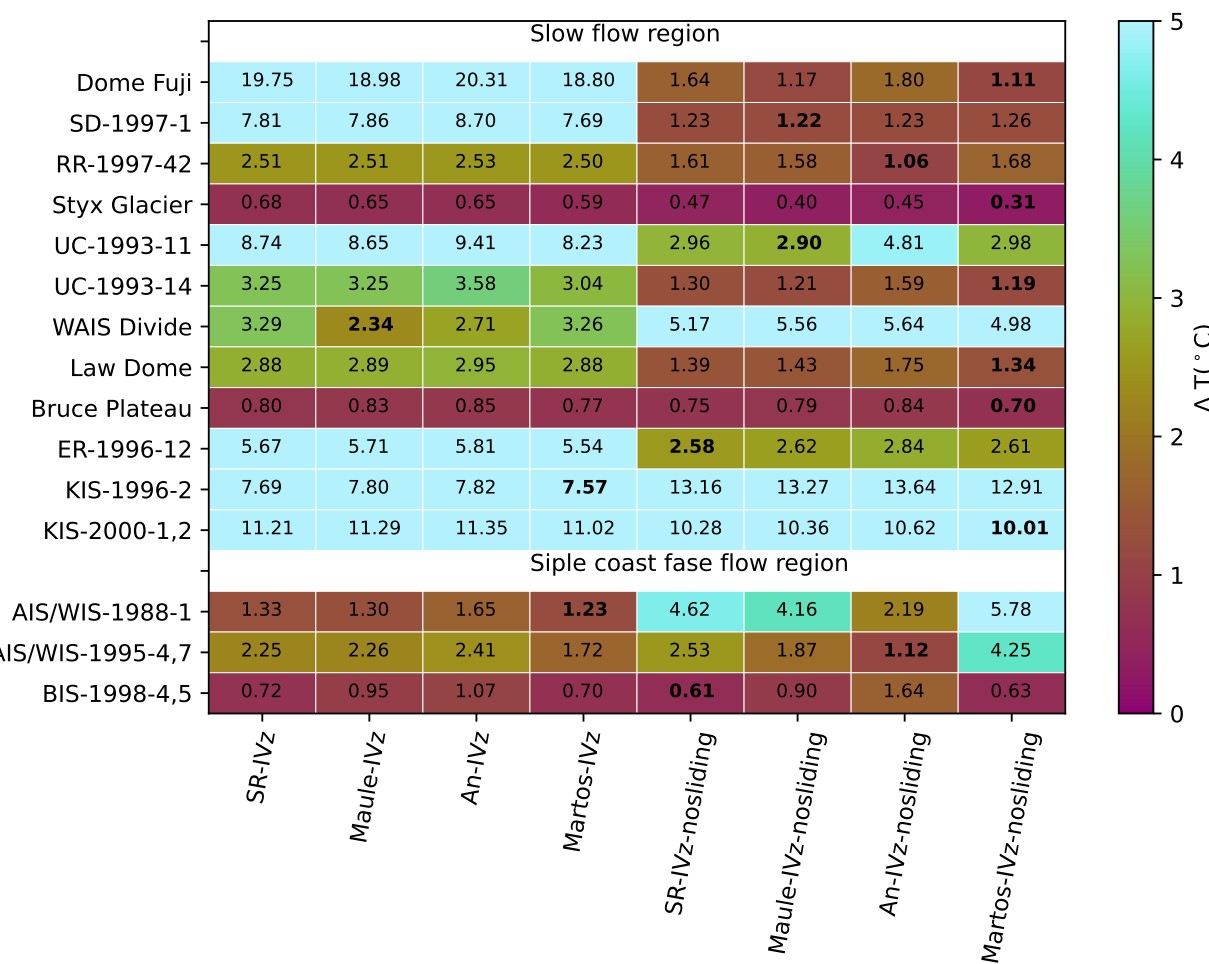

**Figure 3.** Weighted absolute misfit between observed and modeled borehole temperatures according to each experiment. The absolute temperature misfit is truncated over $5^\circ$C.

| Borehole Name | IVz | | | | | IVz-nosliding | | | | |
|---|---|---|---|---|---|---|---|---|---|---|
| | SR | Maule | An | Martos | mean | SR | Maule | An | Martos | mean |
| Slow flow region | | | | | | | | | | |
| Dome Fuji | -0.14 | -0.14 | -0.14 | -0.14 | -0.14 | -0.01 | -0.01 | -0.00 | -0.01 | -0.01 |
| SD-1997-1 | -0.40 | -0.41 | -0.39 | -0.41 | -0.40 | -0.08 | -0.08 | -0.04 | -0.08 | -0.07 |
| RR-1997-42 | -0.13 | -0.13 | -0.12 | -0.13 | -0.13 | -0.09 | -0.09 | -0.05 | -0.10 | -0.08 |
| Styx Glacier | -0.29 | -0.31 | -0.28 | -0.33 | -0.30 | -0.07 | -0.08 | -0.07 | -0.08 | -0.08 |
| UC-1993-11 | -0.19 | -0.21 | -0.18 | -0.22 | -0.20 | 0.01 | -0.00 | -0.01 | 0.02 | 0.00 |
| UC-1993-14 | -0.43 | -0.43 | -0.41 | -0.44 | -0.43 | -0.30 | -0.36 | -0.41 | -0.34 | -0.35 |
| WAIS Divide | 0.02 | 0.03 | 0.03 | 0.02 | 0.03 | 0.17 | 0.18 | 0.18 | 0.16 | 0.17 |
| Law Dome | -1.49 | -1.51 | -1.46 | -1.55 | -1.50 | -0.02 | -0.02 | 0.01 | -0.03 | -0.01 |
| Bruce Plateau | -5.19 | -5.25 | -5.19 | -5.34 | -5.24 | -2.65 | -2.73 | -2.82 | -2.66 | -2.71 |
| ER-1996-12 | -0.23 | -0.23 | -0.23 | -0.23 | -0.23 | -0.22 | -0.23 | -0.22 | -0.22 | -0.22 |
| KIS-1996-2 | 0.04 | 0.05 | 0.05 | 0.04 | 0.05 | -0.06 | -0.07 | -0.07 | -0.07 | -0.06 |
| KIS-2000-1,2 | 0.29 | 0.28 | 0.30 | 0.27 | 0.28 | 0.03 | 0.03 | 0.03 | 0.02 | 0.03 |
| Siple coast fast flow region | | | | | | | | | | |
| AIS/WIS-1988-1 | 0.25 | 0.25 | 0.26 | 0.22 | 0.25 | 0.07 | 0.16 | 0.28 | 0.01 | 0.13 |
| AIS/WIS-1995-4,7 | 0.35 | 0.35 | 0.37 | 0.34 | 0.35 | 0.26 | 0.32 | 0.33 | 0.31 | 0.31 |
| BIS-1998-4,5 | 2.48 | 2.82 | 1.55 | 3.44 | 2.57 | -2.59 | -2.79 | -2.58 | -3.11 | -2.76 |

**Table 3.** Depth-averaged vertical velocity for each experiment at each borehole. Positive values indicate upward advection.

The borehole of Styx Glacier is a shallow ice core limited to 210.5 m (Yang et al., 2018), even though the ice thickness from ice penetrating radar is approximately 550 m (Hur, 2013). Both IVz and IVz-nosliding groups display similar average misfit values of ∼0.64°C and ∼0.40°C, which show good agreement with the observed temperature profile. The thermal model results suggest that we do not reach the melting point at the borehole of Styx Glacier in all experiments.

The measured UC borehole temperature profile displays a relatively high basal temperature gradient compared to the other adjacent boreholes, such as AIS/WIS boreholes (Engelhardt, 2004b). The mean GHF in the UC region is approximately 81.4 mW m$^{-2}$ for SR, 86.5 mW m$^{-2}$ for Maule, 62.8 mW m$^{-2}$ for An, and 95.6 mW m$^{-2}$ for Martos. The current modeled temperature profiles at UC-1993-11 and UC-1993-14 agree well with the measured temperature regardless of the choice of GHFs. The misfit value for the modeled and observed temperatures from the IVz-nosliding group is lower than that of the IVz group. In addition, the misfit of UC-1993-14 for IVz-nosliding is lower than that of UC-1993-11 (Figure 3). UC-1993-14 is located in a slow region; however, UC-1993-11 is adjacent to the shear margin of the AIS ice stream, which induces a sharp transition in the basal velocity constraints for the IVz-nosliding group where the ice velocity crosses 10 m yr$^{-1}$. While the IVz-nosliding group captures better the observed temperature profiles for UC-1993-14, it is not the case for UC-1993-11.

The modeled basal temperature at the WAIS Divide reaches the pressure melting point only for the SR and Martos IVz groups. The GHF is approximately 112.6 mW m$^{-2}$ for SR, and 141 mW m$^{-2}$ for Martos; these values are higher than those

of the other two GHF datasets, which are 60.3 mW m$^{-2}$ for Maule and 68.9 mW m$^{-2}$ for An. The basal melting rate of the IVz-nosliding group is 7.9 mm yr$^{-1}$ for SR, 2.5 mm yr$^{-1}$ for Maule, 3.4 mm yr$^{-1}$ for An, and 10.9 mm yr$^{-1}$ for Martos. GHF estimations in previous studies are 113.3$\pm$16.9 mW m$^{-2}$ from Talalay et al. (2020) and 90.5 mW m$^{-2}$ from Mony et al. (2020). The thickness at WAIS Divide is 3455 m (WAIS Divide Project Members, 2013). However, the drilling depth is 3405 m (Slawny et al., 2014), and does not reach the bed, so we do not know the rate of basal melting. According to Talalay
et al. (2020), the estimated basal temperature at WAIS Divide reaches the pressure melting point, and the basal melting rate is about 3.7 $\pm$ 1.7 mm yr$^{-1}$. All experiments show reasonably good agreement in terms of the shape of the observed borehole temperature profile at WAIS Divide regardless of the choice of GHF. The average misfit value of the borehole temperature for IVz is 2.90°C, and is better than that of IVz-nosliding (Figure 3).

At Law Dome, the misfit between the observed and modeled temperatures is 2.9°C and 1.5°C for the IVz and IVz-nosliding
groups, respectively (Figre 3). A primary difference between IVz and IVz-nosliding is the depth-averaged vertical velocity, which of value is -1.5 m yr$^{-1}$ for IVz group and -0.1 m yr$^{-1}$ for IVz-nosliding (Table 3). In the Law Dome case, we confirm that the use of IVz-nosliding improves the model's vertical temperature profile (Figure 2).

The observed ice velocity at Bruce Plateau is 49 m yr$^{-1}$ according to MEaSUREs version 2 Rignot (2017), which is higher than the previously reported value of 10 $\pm$ 4 m yr$^{-1}$ (Zagorodnov et al., 2012). We find that none of the modeled thermal
profiles can reproduce the upper part of the observed ice temperature that captured the colder surface temperature of past climate (Zagorodnov et al., 2012). The mean vertical velocity for the IVz group is -5.24 m yr$^{-1}$, and -2.71 m yr$^{-1}$ for the IVz-nosliding group; these values indicate high vertical advection toward the bottom.

Except for ER-1997-12, neither IVz nor IVz-nosliding group capture the observed temperature profiles at the KIS boreholes. All modeled temperature profiles exhibit a convex shape (Figure 2). At ER-1997-12, the mean misfit between the modeled and
observed temperature is 2.7°C for the IVz-nosliding group and 5.7°C for the IVz group (Figure 3).

The AIS/WIS and BIS boreholes are located in Siple coast fast flow regions where the ice velocities are 365 m yr$^{-1}$ for AIS/WIS-1991-1, 379 m yr$^{-1}$ for AIS/WIS-1995-4,7, and 376 m yr$^{-1}$ for BIS-1998-4,5 from MEaSUREs version 2 (Rignot, 2017). In these regions, both IVz and IVz-nosliding allow for basal sliding, and the vertical velocities calculated for the IVz-nosliding and IVz groups are not significantly different, as expected. The modeled temperature profiles for IVz and IVz-
nosliding show similar results. The average misfit value of the IVz group is 1.38°C for AIS/WIS-1988-1, 2.16°C for AIS/WIS-1995-4,7, and 0.86°C for BIS-1998-4,5 (Figure 3). The misfit value of the modeled and observed temperatures at BIS is lower than that of AIS/WIS. The primary difference between the BIS and AIS/WIS regions is that the bed geometry in the BIS region is constructed using a mass conservation approach, which relies on the equation of ice incompressibility. In contrast, the bed geometry in the AIS/WIS region was constructed using the stream diffusion method, which produces bed topography similar
to kriging (Figure S5). This suggests that enhancement in the quality of the geometry and utilizing the mass conservation method in the Siple coast fast flow regions would improve the estimation of the vertical velocity by the IVz equation with sliding as well as the overall performance of the thermal model. The AIS/WIS-1995-4,7 borehole is located at the center of the ice stream, whereas AIS/WIS-1988-1 is relatively near the margin of the ice stream. Although the bed geometry at AIS/WIS

was constructed using the kriging method, IVz reproduces the temperature profile reasonably well at the center of fast ice flow
regions.

### 3.3   Subglacial conditions

Figure 4a and 4b show the mean and standard deviation of the basal temperature distribution for the eight experiments. The
mean basal temperature at the main ice trunk, where the ice primarily discharges into the ocean, reaches the ice pressure
melting point. The standard deviation of the basal temperature is higher in the internal ice compared to the peripheral regions.
In the case of IVz-nosliding, constraining the basal velocity to zero in slow flow regions leads to a warmer basal temperature
distribution compared to the IVz group. In slow flow regions, the basal temperature of the IVz group shows a notable difference
depending on the choice of GHF. The modeled basal temperatures in the Maule and Martos experiments, which have higher
mean GHF values (Table 4), are warmer than those in SR and An experiments, as expected. The mean GHF of An is the lowest
compared to the other GHFs, and therefore, the basal temperature at each borehole modeled with the An GHF is lower than
those of the other GHFs.

All the experiments generally indicate that most of the regions experiencing basal melting are concentrated in fast flow
regions, where basal frictional heat is significant and provides enough heat for the ice to reach the pressure melting point
(Figure 4). Since IVz-nosliding displays lower vertical advection than that of IVz, the basal temperature of the IVz-nosliding
group in slow flow regions is warmer than that of IVz (Figure 4c-j).

The mean total grounded ice melting volume is 26.62 Gt yr$^{-1}$ for the IVz group, and 29.77 Gt yr$^{-1}$ for the IVz-nosliding
group (Table 4). The total grounded ice melting volume for the IVz-nosliding group is 3.15 Gt yr$^{-1}$ higher than that of IVz.
The surplus of the mean total basal melting volume of the IVz-nosliding group is supplied by 1.89 Gt yr$^{-1}$ (60%) and 1.08 1.26
Gt yr$^{-1}$ (40%) in the slow and fast flow regions, respectively. The total melting fraction of the grounded ice, which represents
the grounded ice melting area, is 53.60% for the IVz group, and 61.54% for the IVz-nosliding group (Table 4). Each basin
displays significant differences in terms of the grounded ice melting volume depending on the GHF source. Martos provides a
high GHF along the West Antarctic Rift System, and therefore, Martos' total grounded ice melting volume in basins 8 and 10
has the highest values of 9.33 Gt yr$^{-1}$ and 10.28 Gt yr$^{-1}$ for IVz and IVz-nosliding, respectively. Basin 5, including Totten,
Moscow University, and Holmes Glacier, shows the highest total grounded ice melting volume in East Antarctica, excluding
basins 8 and 15. The GHF from An, which is the lowest value among all GHFs, shows the lowest total grounded ice melting
volume.

## 4   Discussion

Previous studies that have successfully reproduced borehole temperature profiles using one-dimensional thermal analytical
solutions have been limited to slow flow regions (Joughin et al., 2003; Mony et al., 2020; Talalay et al., 2020). These studies
have demonstrated good agreement between modeled and observed temperatures, which is expected given their simplicity and
tunability of the analytical solutions. One important tunable parameter is the analytical vertical velocities, which rely on ice

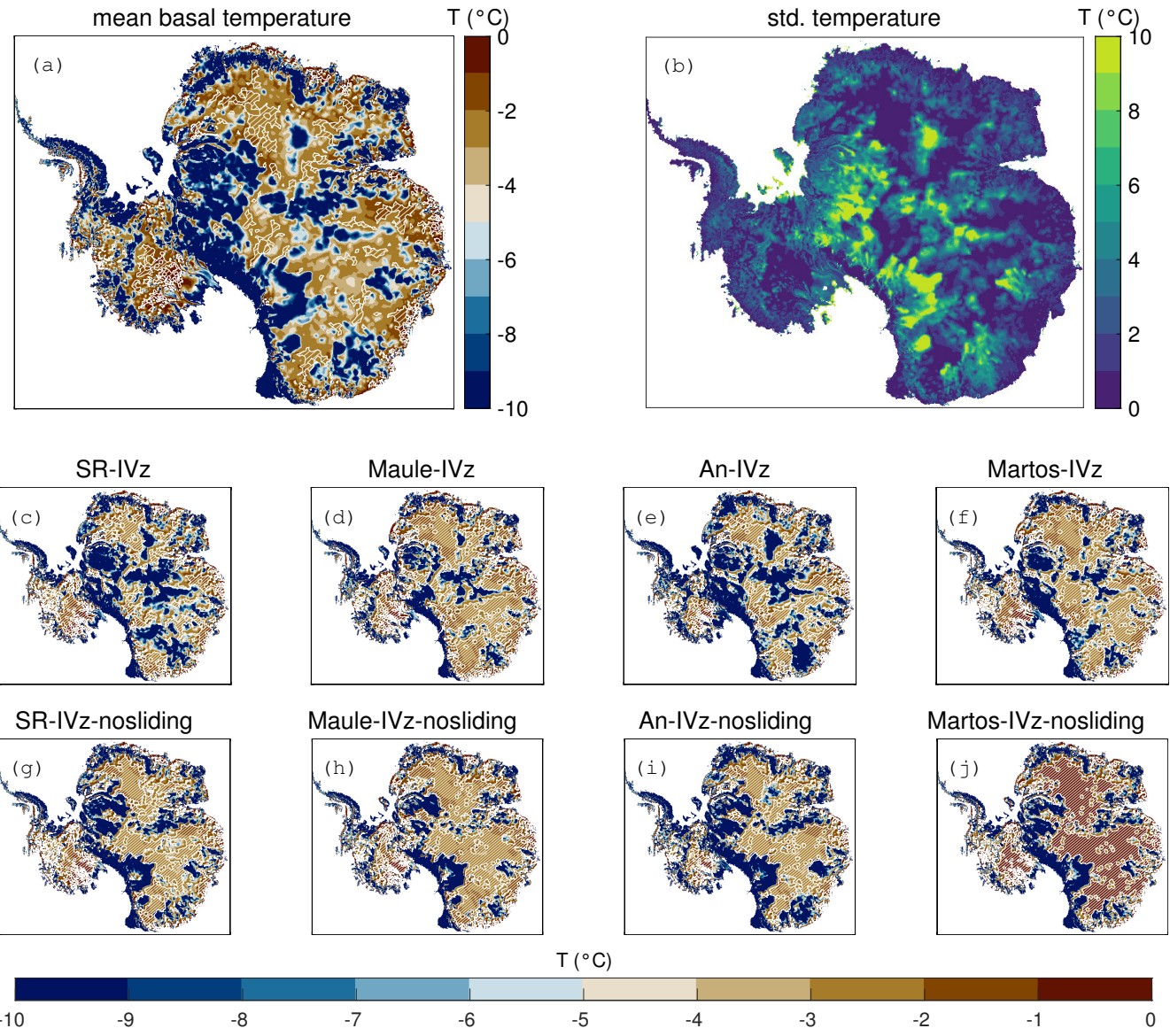

**Figure 4.** (a) Mean and (b) standard deviation of the basal temperature distribution from eight experiments. (c-j) Basal temperature distribution for each experiment. The temperature legend is truncated below -10 °C. White slash line region indicates that the basal temperature of ice reaches the pressure melting point.

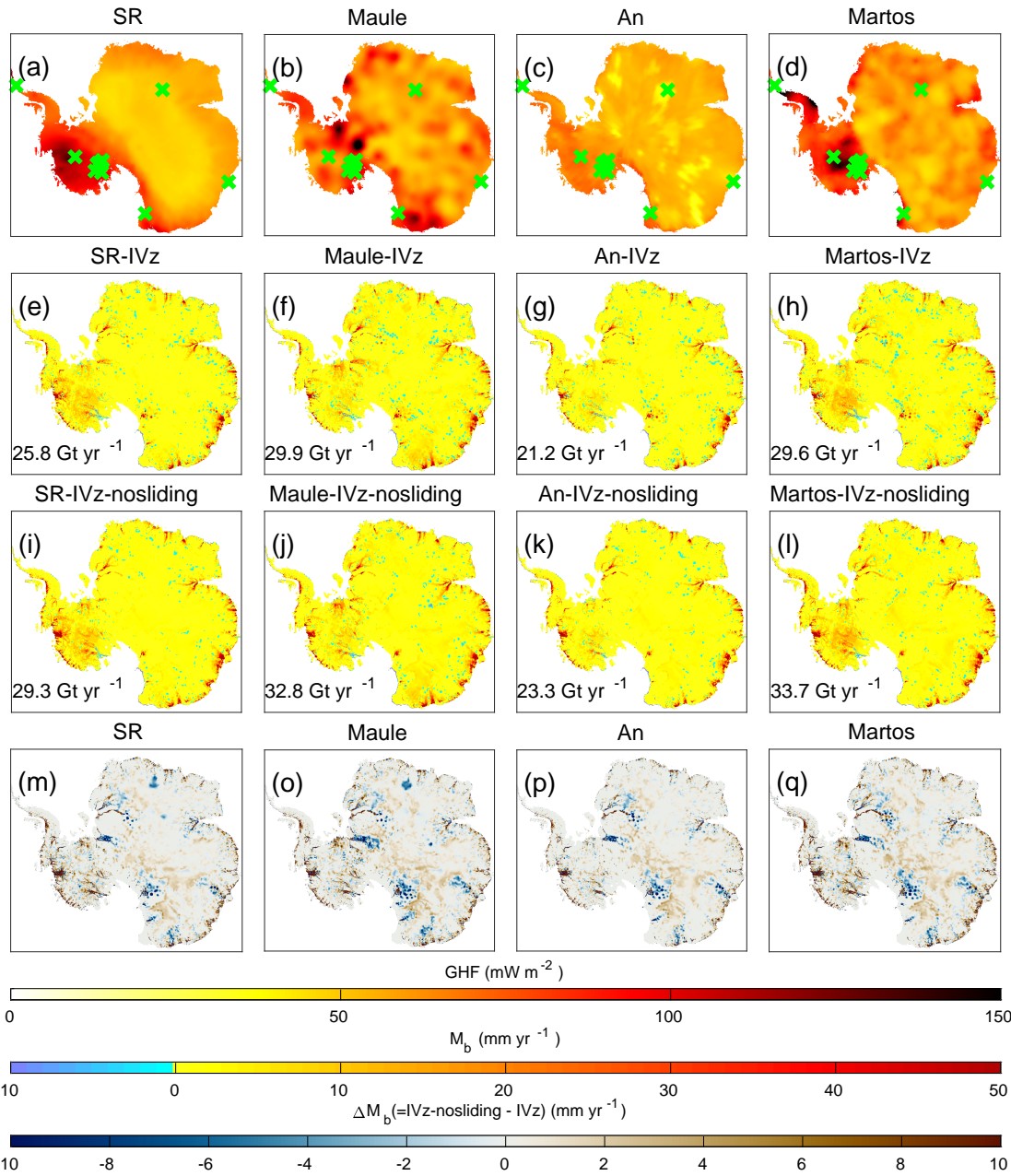

**Figure 5.** Upper panels (a-d) are the geothermal heat flux distributions of each source. Middle panels (e-l) are the basal melting rate distributions, with the value at the bottom left indicating the total grounded ice melting volume for each experiment. The basal melting rate exceeding 50 mm yr$^{-1}$ is truncated. Lower Panels (m-q) are difference in basal melting rate between IVz-nosliding and IVz for each geothermal heat flux. A green cross dot on the geothermal heat flux map indicates the borehole location. The color map for difference in basal melting rates is from Crameri et al. (2020).

| | Vertical velocity | Grounded ice melting volume (Gt yr$^{-1}$) | | | | | | | |
| | | IVz | | | | IVz-nosliding | | | |
| | GHF | SR | Maule | An | Martos | SR | Maule | An | Martos |
| Basin id | | | | | | | | | |
|---|---|---|---|---|---|---|---|---|---|
| | 1 | 1.16 | 1.36 | 0.82 | 1.07 | 1.20 | 1.41 | 0.93 | 1.36 |
| | 2 | 0.71 | 0.89 | 0.72 | 0.93 | 0.90 | 1.11 | 0.87 | 1.19 |
| | 3 | 1.64 | 2.27 | 1.60 | 2.18 | 1.63 | 2.38 | 1.64 | 2.30 |
| East Antarctica | 4 | 1.65 | 2.46 | 1.64 | 2.13 | 2.05 | 2.92 | 1.94 | 2.64 |
| | 5 | 3.65 | 4.40 | 3.68 | 4.73 | 4.15 | 4.78 | 3.87 | 5.27 |
| | 6 | 1.95 | 3.32 | 1.49 | 1.85 | 2.10 | 3.44 | 1.52 | 2.00 |
| | 7 | 0.39 | 0.63 | 0.26 | 0.25 | 0.44 | 0.63 | 0.30 | 0.31 |
| | 16 | 0.21 | 0.41 | 0.18 | 0.35 | 0.30 | 0.51 | 0.25 | 0.46 |
| Ross ice shelf | 8 | 3.98 | 3.53 | 2.65 | 5.04 | 4.49 | 3.84 | 2.90 | 5.50 |
| | 9 | 1.06 | 0.82 | 0.80 | 0.84 | 1.41 | 1.12 | 1.07 | 1.21 |
| West Antarctica | 10 | 5.09 | 3.59 | 3.44 | 4.29 | 5.78 | 3.91 | 3.77 | 4.78 |
| | 11 | 0.33 | 0.23 | 0.25 | 0.27 | 0.46 | 0.33 | 0.34 | 0.41 |
| | 12 | 0.76 | 0.81 | 0.72 | 0.90 | 1.00 | 1.05 | 0.89 | 1.18 |
| Antarctic Peninsula | 13 | 0.03 | 0.02 | 0.02 | 0.04 | 0.02 | 0.01 | 0.01 | 0.04 |
| | 14 | 0.00 | 0.01 | 0.00 | 0.02 | 0.01 | 0.01 | 0.00 | 0.03 |
| Ronne-Filchner ice shelf | 15 | 3.14 | 5.12 | 2.93 | 4.73 | 3.35 | 5.38 | 2.94 | 5.01 |
| Total grounded ice melting volume (Gt yr$^{-1}$) | | 25.78 | 29.86 | 21.21 | 29.64 | 29.29 | 32.84 | 23.26 | 33.70 |
| Grounded ice melting fraction (%) | | 48.29 | 61.70 | 45.39 | 59.01 | 59.44 | 66.62 | 54.53 | 65.58 |
| Mean GHF (mW m$^{-2}$) | | 60.78 | 65.61 | 54.66 | 65.49 | 60.78 | 65.61 | 54.66 | 65.49 |

**Table 4.** Grounded ice basal melting volumes of eight experiments at each basin (Figure 1) as well as the total grounded ice melting volume and the total grounded melting fraction corresponding to each experiment.

surface mass balance (Hindmarsh, 1999; Joughin et al., 2003; Talalay et al., 2020). The choice of vertical velocity is a key factor in reproducing borehole temperature profiles. Uncertainties in the GHF have also been identified as a major factor in reproducing observed borehole temperature profiles (Talalay et al., 2020; Mony et al., 2020). On the other hand, some other studies have shown that uncertainties in the GHF have little influence on model performance in terms of ice dynamics (Larour et al., 2012a; Smith-Johnsen et al., 2020a), and simulating future projections (Schlegel et al., 2018; Smith-Johnsen et al., 2020b; Seroussi et al., 2013). Therefore, to test other factors, such as different GHFs and vertical velocities, that may affect the calculation of borehole temperatures, we use a three-dimensional thermo-mechanical model in order to account for both horizontal and vertical advection. We compare our calculated temperatures to observed borehole profiles in both fast and slow flow regions.

In slow flow regions, we find that IVz-nosliding experiments show a reasonably good agreement with the observed borehole temperature profiles. However, the three-dimensional thermal model occasionally estimates convex temperature profiles, which are not consistent with the observations, such as KIS-1996-2 and KIS-2000-1,2 boreholes. Compared to other boreholes, the ice velocities at KIS and ER gradually decrease from upstream to downstream, and coincide with the presence of a basal ridge (Price et al., 2001; Ng and Conway, 2004) (see also Figure S2). Furthermore, the subglacial hydrology system at WAIS discharging to the Ross Ice Shelf has been explored using magnetotelluric, passive seismic data, and drilling borehole (Fisher et al., 2015; Priscu et al., 2021; Gustafson et al., 2022). The deceleration of tributaries at KIS and ER is attributed to water-piracy hypothesis (Alley et al., 1994) or removal of basal water contributing to the loss of lubrication (Tulaczyk et al., 2000; Bougamont et al., 2003). In model experiments, Bougamont et al. (2015) revealed changes in the tributaries at KIS and ER using a plastic till deformation friction law including simple subglacial hydrology model. In contrast, we employ the Budd type friction law and assume the effective pressure fully connected to ocean part, not including changes in the effective pressure. The variation in effective pressures also changed the basal ice velocity in Budd type friction law. In addition, a selection of other types of friction law, including Weertman (Weertman, 1974), Schoof (Schoof, 2005), and Coulomb (Tsai et al., 2015) types, also influences the initialization and future fate of ice (Brondex et al., 2017, 2019). Further investigation is required, such as the application of other types of friction laws or initialization with paleo spin-up, to better understand temperature profiles.

In Siple coast fast flow regions, Joughin et al. (2004) utilized a thermal model with vertical velocity derived from an analytical solution, which reproduced the observed borehole temperature profile of BIS-1998-4,5 with good agreement (UpD in Joughin et al. (2004)). Here, we also find that the modeled temperature using a vertical velocity based on the equation of incompressibility without any constraint or tunable parameter also agrees well with the observed temperatures in this sector.

The total grounded ice melting volume for both the IVz and IVz-nosliding groups falls within the range reported by previous studies. It is lower than 65 Gt $yr^{-1}$ from Pattyn (2010) and higher than 14.7 Gt $yr^{-1}$ from Llubes et al. (2006), which of value is converted to volume from total ice volume of 16 $km^3$ $yr^{-1}$ in ice equivalent. In the study by Joughin et al. (2009), they adopted a homogeneous GHF value of 70 mW $m^{-2}$, which is similar to the mean GHFs from Maule, 66.95 mW $m^{-2}$, and An, 67.15 mW $m^{-2}$ at basin 10, which includes Pine Island and Thwaites Glaciers (see basin in Figure 1). However, the total grounded ice melting mass estimated by Joughin et al. (2009), 5.2 Gt $yr^{-1}$, is higher than that of IVz group (average value of Maule and An), 3.5 Gt $yr^{-1}$, and IVz-nosliding group (average value of Maule and An), 3.8 Gt $yr^{-1}$. Joughin et al. (2009) assumed that the vertical velocity varied linearly from the surface mass balance at the surface to zero at the bed in fast flow regions, and they relied on an analytical solution similar to the one presented by Hindmarsh (1999) in slow flow regions (Dansgaard and Johnsen, 1969; Dahl-Jensen et al., 1999). Since Joughin et al. (2009) used an analytical solution that underestimated the magnitude of the vertical velocity compared to the vertical velocity obtained from the equation of ice incompressibility, it results in overestimation of the total grounded ice melting volume in basin 10.

Thermal models have been used to reconstruct the thermal regime of ice and estimate the melting volume beneath grounded ice. Regarding the advection term in the thermal model, horizontal ice velocity is estimated with HO or FS models, while the vertical velocity is recovered with the ice incompressibility. Under kriging-based geometry, the vertical velocity in fast flow region does not coincide with physical property. In contrast, state-of-the art bed geometry, such as BedMachine (Morlighem

et al., 2017, 2020), is generated with the mass conservation, which of equation is based on ice incompressibility. We confirm that using the equation of ice incompressibility to reconstruct the ice vertical velocity provides a viable way of computing temperature profiles that exhibit good agreement with observations in Siple coast fast flow regions, such as the BIS. Given that the geometry of other fast flow regions, such as Thwaites Glacier, is generated using the mass conservation method (Morlighem et al., 2011, 2020), therefore, we expect that this study provides a method to generate reliable temperature profiles. Note that

the good agreement in modeled temperature at fast flow region, not only Siple coast fast flow region, does not guarantee the magnitude of basal melting volume because the basal melting volume at fast flow region is associated with the frictional heat. However, at slow flow region, the basal temperature is mainly affected by the GHF and the vertical advection, rather than the low frictional heat. Therefore, it is noteworthy that the basal melting rate produced using IVz-nosliding in slow flow regions would be reliable.

We find that the impact of using different GHF fields has only a modest influence on the ice temperature field and the total grounded ice basal melting volume. Under these circumstances, our results reveal that the shapes of the borehole temperature profile are less sensitive to the current estimated GHFs than previously reported. It is also worth noting that the initialization with the GHF from An results in underestimated basal temperatures and a lower total grounded ice melting volume due to an excessively low GHF value compared to other datasets.

The IVz-nosliding experiment has the advantage of better simulating the vertical temperature profiles in slow flow regions compared to IVz. However, it tends to produce large discrepancies between modeled and observed surface ice velocities (Figure S3). For instance, the An-IVz-nosliding thermal model experiments exhibit the largest misfits in ice velocity among all the experiments, as the lowest value of average GHF leads to relatively high ice rigidity that perturbs the ice flow in the slow flow regions. In contrast, IVz experiment shows relatively smaller misfit values in surface ice velocity because sliding

compensates for the underestimated internal deformations in the slow flow region. In general, we find that IVz leads to a higher depth-averaged ice rigidity compared to IVz-nosliding in slow regions due to presence of colder ice temperatures (Figure S4). Higher ice rigidity causes ice to deform less vertically, through vertical shear, and the surface ice velocity with no-sliding cannot reproduce the observed surface velocities. In other words, the surface ice velocity of IVz-nosliding shows a larger ice velocity misfit compared to that of the IVz group, because the basal velocities are constrained to zero and cannot compensate

for the high velocity misfit. Furthermore, the adoption of no-sliding in specific regions results in a sharp transition zone in ice rigidity, B. This occurs because the basal velocity near the transition zone does not smoothly changed from no-sliding to sliding (Figure S4). Therefore, additional work is required to address and resolve the smooth transition between no-sliding and sliding.

In slow flow regions, a competition between vertical diffusion and advection determines the shape of the temperature profiles

and the bottom temperatures. In the IVz experiments, the boundary condition for basal vertical velocity is recovered with the gradient of the bed geometry and the basal melting volume. This approach provides relatively high vertical velocities in slow flow regions. The vertical velocities are not always in agreement with the analytical expression of vertical velocities assuming a stationary bed and no-sliding. As the depth-averaged vertical velocity of IVz is higher than that of IVz-nosliding, cold surface temperatures can be more effectively transferred deeper into the ice column.

The surface temperature of ice would be one of factors to consider the boundary condition of thermal model. While ERA5 (Hersbach et al., 2023), RACMO2.3p2 forced with ERA5 (van Wessem et al., 2023), and MERRA2 (Global Modeling and Assimilation Office (GMAO), 2015) are the recent reanalysis datasets, they display some discrepancies between the climatological mean 2-m air temperature (1980-2018) and observed surface temperature at each borehole (Figure S6). For the comparison with different version of ECMWF (European Centre for Medium-Range Weather Forecasts) reanalysis data, we

perform experiments using the same manner, utilizing 2-m air temperature from ERA5. These results display no significant differences compared to experiments using ERA-Interim (Figure S7). However, in case of SD, RR, and AIS/WIS (only for the IVz-nosliding case), they display slight discrepancies in surface temperature leading to shifts in the modeled temperature profiles. In fact, the improvement in surface temperature and the accurate correction would bring the modeled temperatures into closer agreement with observations.

Finally, borehole temperatures have a long-term memory of past climate air temperatures and are a good proxy for reconstruction over a few hundred years or longer using inverse modeling (Nagornov et al., 2001; Zagorodnov et al., 2012). This history is not accounted for in this study as we assumed thermal steady state using current climatological information. Despite this strong limitation, we find that this approach provides temperature profiles that are in good agreement with observations.

## 5    Conclusions

In this study, we used a three-dimensional thermo-mechanical model of Antarctica with different sources of GHF and vertical velocity fields to reproduce different thermal states of the Antarctic ice sheet, and we compared the results to 15 in situ measured borehole temperature profiles in slow and fast flow regions. Comparing the modeled to measured borehole temperature profiles, we confirm that the vertical ice velocity based on the equation of incompressibility (IVz) is suitable for fast flow regions, such as BIS, where the bed geometry is constructed with using the mass conservation method, while an IVz that ignores basal sliding

(IVz-nosliding) performs better in slow flow regions. Our results show that the vertical temperature profile is more sensitive to the vertical velocity. In addition, the basal conditions, such as temperature and melting rate, are both sensitive to both GHF and the vertical velocity field. The total grounded ice melting volume and basal temperature are proportional to the magnitude of the average GHF values for the same vertical velocity method. Finally, constraining the basal velocity to zero in slow flow regions is a reasonable assumption and leads to a more realistic temperature profile.

*Code and data availability*. ISSM is open source and can be download at https://issm.jpl.nasa.gov. Law Dome temperature profile by Van Ommen et al. (1999) is available at https://data.aad.gov.au/metadata/records/lawdome_borehole_temp_1987. Dome Fuji temperature profile is available at Hondoh et al. (2002). Styx Glacier borehole temperature profile by Yang et al. (2018) is obtained with personal communication. Bruce Plateau temperature profile is available at Zagorodnov et al. (2012). WAIS Divide borehole temperature by Cuffey and Clow (2014) is available http://dx.doi.org/10.7265/N5V69GJW. SD, RR, UC, ER, KIS, AIS/WIS, BIS borehole temperature by Engelhardt (2004b)

are available at http://dx.doi.org/10.7265/N5PN93J8. GHF map by Shapiro and Ritzwoller (2004) and Maule et al. (2005) are available at ALBMAP v1.0 (http://doi.pangaea.de/10.1594/PANGAEA.734145). GHF map by An et al. (2015) is available at http://www.seismolab.org/

model/antarctica/lithosphere/index.html. GHF map by Martos et al. (2017) is available at https://doi.pangaea.de/10.1594/PANGAEA.882503. 2-m air temperature by Dee et al. (2011) is available at https://www.ecmwf.int/en/forecasts/datasets/reanalysis-datasets/era-interim. Ice velocity map by Rignot (2017) is available at https://nsidc.org/data/nsidc-0484/versions/2. Bed geometry of Antarctica by Morlighem et al. (2020) is available at https://nsidc.org/data/NSIDC-0756/versions/1. Surface elevation map by Howat et al. (2019) is available at https://www.pgc.umn.edu/data/rema/). The results of gridded basal temperature field are available at KDPC (Korea Polar Data Center) (https://dx.doi.org/doi:10.22663/KOPRI-KPDC-00002216.3).

*Author contributions.* IW Park designed and performed experiments with inputs from EK Jin, M. Morlighem, and KK Lee. All authors participated in the writing of the manuscript.

*Competing interests.* The authors declare that they have no conflict of interest.

*Acknowledgements.* The authors are grateful to the editor, Dr. Benjamin Smith, for handle handling our article and to the two reviewers, Dr. Tyler Pelle and anonymous, for their helpful comments on the manuscript. This research was supported by Korea Institute of Marine Science and Technology Promotion (KIMST) funded by the Ministry of Oceans and Fisheries (RS-2023-00256677; PM23020) and Korea Polar Research Institute (KORPI). MM was funded by the PROPHET project, a component of the International Thwaites Glacier Collaboration (ITGC), with support from National Science Foundation (NSF: Grant #1739031).

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
