# Peer review of "Impact of boundary conditions on the modeled thermal regime of the Antarctic ice sheet"

_The Cryosphere, 2023_

## Referee Comment (RC1)

**Title**: Impact of boundary conditions on the modeled thermal regime of the Antarctic ice sheet
**Journal**: *The-Cryosphere*
**Reviewer**: Tyler Pelle (Scripps Institution of Oceanography, tpelle@ucsd.edu)

**Overview**:
Park et al. present an in-depth analysis of how varying geothermal heat flux fields and vertical ice velocity initializations impact the modeled thermal regime of the Antarctic Ice Sheet (AIS) via comparison to 15 borehole measurements. Using the three-dimensional Ice-sheet and Sea-level System Model to provide 8 modeled thermal AIS states (4 geothermal heat flux fields and 2 vertical ice velocity initializations), Park et al. find that varying vertical ice velocities have the greatest impact on the modeled thermal state and that traditional means of inferring vertical ice velocity perform well in fast flowing regions.

Overall, I find that the paper is very well written and the results will be of wide interest to those within the glaciological community. This work constitutes an important step forward in our understanding of how ice sheet thermal models perform against available borehole measurement and which initialization processes drive the thermal solution. I do have a few general comments about that paper that I would like to see addressed, but these are mostly minor and should be relatively easy for the authors to fix. In particular, I am a bit worried that the conclusion that "GHFs have little influence on the variance in basal temperature fields and grounded ice melting rate compared to the vertical velocities" is not well supported by the work (see line comment L343). I also would like to see a bit more explanation about limitations of the ice sheet model and how it is initialized. Otherwise, most of the remaining comments are grammatical or based on small changes I would like to see to figures (most of which are very well constructed). I think this work would make a wonderful contribution to *The-Cryosphere* and I would like to see it published after addressing my minor comments.

**General Comments**:
- **Abstract**: Your manuscript is full of really wonderful conclusions that didn't make it into the abstract! For instance, a lot of your results pertain to modeled grounded ice melting rates and how varying spatial distributions of GHF impact this. In addition, you also highlighted that bed topography from mass conservation improved the performance of the thermal model over other methods that are less constrained by data. While it is up to you which results you would like to highlight and I do appreciate that you kept the abstract very straight forward, I think a lot of really great results are buried in the paper and you have the room here to highlight them (same for the conclusions as well).
- **Tense of writing**: When reading, I noticed that you switched between past and present tense a lot. I think the standard is to use the present tense. In the line comments, I tried to point out a few instances of when you used past tense, but I definitely did not catch all of the instances.

- **Assumptions in the ice sheet model**: Will the choice of a Budd sliding law impact the simulated thermal structure of the AIS? Same for the assumption that the effective pressure is equal only to the ice overburden pressure (meaning that you are assuming there is no subglacial water system at the ice-bed interface)? Several studies (e.g. Gustafson et al., 2022) have found a complex subglacial water system underlying the Siple Coast (where many of your borehole measurements are taken), which could certainly impact basal sliding (and thus vertical ice velocities). While I believe that an in-depth analysis of this is beyond the scope of this paper, it would be nice to see perhaps a figure or two in the supplement that show if your modeled thermal AIS states are sensitive to these two assumptions. I would also like to see limitations like this addressed in the discussion section as well.

**Line Comments**:
- L7: Can you be more specific when saying "vertical velocity plays a more important role in the temperature profile than GHF"? Do you mean that the temperature profiles are more sensitive to vertical velocity than GHF?
- L8: " . . . which consists of combining . . .
- L21: "Several important properties, such as ice elevation and surface ice velocity, . . ." " . . . subglacial properties, such as ice temperature and geothermal heat flux, . . ."
- L25: A little confusing wording, perhaps change to: In order to get reasonable estimates of these englacial and subglacial fields, inversion techniques are routinely employed (citations)."
- L28: "geothermal heat flux" should not be capitalized
- L29-30: " . . . and ice dynamics (citations); yet, large uncertainties . . ."
- L36: " . . . better understanding of subglacial and englacial environments . . ."
- Methods: You use past tense here (e.g. "We used ISSM", "We used a 3D HO model", . . .). I mentioned this above, but please try to switch this to present tense when possible.
- L76-78: Which surface ice velocity map did you use to perform the inversions?
- L102:106: In L102, you state that there are three different vertical velocity profiles, but in the experiment description, you only describe 2. Also, in L105, do you mean IVz-nosliding ignores the inferred basal sliding velocities?
- L115: It would be interesting to see a figure showing these four GHF data sets (with overlaid positions of borehole measurements) rather than only giving the AIS-means. I see you do this for the top row of figure 4, maybe reference that here and add borehole locations onto those maps if possible.
- L127: " . . .between temperature measurements along the borehole profile and triangular mesh were not . . ."
- L156: Add a space between sentences.
- L163: Remove double-comma
- L172: Change LVZ to LVz

- Table 3: This table is quite large and it is very difficult to comprehend results from it because there are so many numbers. I am wondering if there is a way you could add shading to the table to show greatest to least misfit (almost like a heatmap). Or you could add highlighting to the lowest misfit for each borehole (replacing the bold text, which does not stand out very much). Alternatively, I'm wondering if this data would be best visualized as a figure rather than a table?
- L181-185: Maybe I am confused, but in figure 2, it seems like the IVz-nosliding group captures the linear shape of the temperature profiles quite well (hence the good match between observations and the dotted lines for the first three profiles). Also, are we interested in the sign of R2, or just its absolute value of? Because while it is true that R2 is larger for IVz-nosliding than IVz, they both seem equally close to 0 when only considering the absolute value. Perhaps a description on how to interpret the R2 value in the methods section would help clarify this.
- L229: Degree symbol is not a superscript in 5.1DegC
- L235-240: This finding about bed topography improving the performance of the thermal model is really great! I know it is not a main point of the paper, but I think it is super important and should be highlighted if possible (maybe in the discussion and/or conclusion sections?).
- L243: Do you know why the vertical profile for Maule-IVz shows such high misfit for the AIS/WIS boreholes, whereas it always seemed fairly similar to the other IVz profiles?
- L246: By peripheral region, do you mean coastal regions? It might be helpful to use a more descriptive word here.
- L257: Perhaps here, it would be better to reference figure 3, where you show the basal temperature fields for each experiment. In figure 3, it is easy to tell that basal temperatures are warmer in IVz-nosliding compared to IVz; however, it is hard to distinguish differences in figure 4 e-l (see figure 4 comments below).
- L262-263: Hard to tell differences in grounded ice melt between GHF sources, see figure-4 comments.
- L340: change "velocitiy" to "velocity"
- Conclusions: I think it might be worth mentioning that varying spatial distributions of GHF did have a large impact on the spatial distribution of grounded ice melting rates across the AIS.
- L343: Here, you say that the effects of different GHFs have little influence on the variance in basal temperature, but I would argue that figure-5 shows the opposite. Comparing figure 5c and 5d (the SR-IVz versus Maule-IVz basal temperature), the area of the ice sheet base that reaches the pressure melting point is much larger for the Maule-GHF than the SR-GHF (especially in East Antarctica). In fact, this difference in the hatched-white area is greater than that when comparing SR-IVz to SR-IVz-nosliding in this same figure, possibly showing that GHF has more of an impact on ice basal

temperature than the vertical ice velocity. As it reads now, I think your conclusions underplay the importance of the GHF field in driving variance in ice basal temperatures.
- L353-355: Fix formatting here.

**Figure Comments**:
- Figure 2: It would be helpful to show visually which borehole measurements were taken in fast-flowing regions. Perhaps in these panels, you could add an asterisk or some identifier.
- Figure 3: This figure is really fantastic!
- Figure 4: Add locations of borehole measurements onto GHF maps if possible. Also, the colormap of the basal melting rate figures is very washed out on the positive side. Perhaps try limiting the colormap to 0.05 m/yr or using a log-scale (with using gray-shading for basal re-freezing since you cannot use log-scale for negative values) to better show regions of grounded ice melt.It could also be interesting to see difference maps in basal melt between respective GHF experiments (e.g. SR-IVz minus SR-IVz-nosliding) as an additional row. Also, it could be helpful to the reader if you include the AIS-integrated grounded ice basal mass balance value in each melt plot so that readers can get a feel for how quantitatively different each result is.
- Figure S2: There are very sharp transitions in B in the IVz-nosliding panel ( see annotated figure below), especially along interior sectors of the EAIS. Does this occur because these are the locations where basal sliding cuts off to 0 m/yr? It might be worth addressing this in the manuscript.

[Figure]

**Citations**:
- Chloe D. Gustafson et al. A dynamic saline groundwater system mapped beneath an Antarctic ice stream. *Science* **376**, 640-644 (2022).

---

## Author Response (AR1)

**Impact of boundary conditions on the modeled thermal regime of the Antarctic ice sheet**
**– Response to Review #1 –**

In-Woo PARK et al.

September 24, 2023

We would like to thank Dr. Tyler Pelle, the reviewer for his insightful and valuable comments, which have helped us to improve our manuscript. We address his remarks below point by point. To clarify the reviewer's comment and its reply, we have used red for the reviewer's comment and black for the replying comment (RC).

**1 Reviewer #1**

**Overview**

Park et al. present an in-depth analysis of how varying geothermal heat flux fields and vertical ice velocity initializations impact the modeled thermal regime of the Antarctic Ice Sheet (AIS) via comparison to 15 borehole measurements. Using the three-dimensional Ice-sheet and Sea-level System Model to provide 8 modeled thermal AIS states (4 geothermal heat flux fields and 2 vertical ice velocity initializations), Park et al. find that varying vertical ice velocities have the greatest impact on the modeled thermal state and that traditional means of inferring vertical ice velocity perform well in fast flowing regions.

Overall, I find that the paper is very well written and the results will be of wide interest to those within the glaciological community. This work constitutes an important step forward in our understanding of how ice sheet thermal models perform against available borehole measurement and which initialization processes drive the thermal solution. I do have a few general comments about that paper that I would like to see addressed, but these are mostly minor and should be relatively easy for the authors to fix. In particular, I am a bit worried that the conclusion that "GHFs have

little influence on the variance in basal temperature fields and grounded ice melting rate compared to the vertical velocities" is not well supported by the work (see line comment L343). I also would like to see a bit more explanation about limitations of the ice sheet model and how it is initialized. Otherwise, most of the remaining comments are grammatical or based on small changes I would like to see to figures (most of which are very well constructed). I think this work would make a wonderful contribution to The-Cryosphere and I would like to see it published after addressing my minor comments.

**General Comments:**

- Abstract: Your manuscript is full of really wonderful conclusions that didn't make it into the abstract! For instance, a lot of your results pertain to modeled grounded ice melting rates and how varying spatial distributions of GHF impact this. In addition, you also highlighted that bed topography from mass conservation improved the performance of the thermal model over other methods that are less constrained by data. While it is up to you which results you would like to highlight and I do appreciate that you kept the abstract very straight forward, I think a lot of really great results are buried in the paper and you have the room here to highlight them (same for the conclusions as well).

RC: Thank you for this comment. As mentioned by the reviewer, we have modified the abstract to emphasize the impact of GHF on the modeled grounded ice melting rates, and the use of mass-conservation-based bed topography data in improving the performance of the thermal model.

Modified abstract:

A realistic initialization of ice flow models is critical for predicting future changes in ice sheet mass balance and their associated contribution to sea level rise. The initial thermal state of an ice sheet is particularly important as it controls ice viscosity and basal conditions, thereby influencing the overall ice velocity. Englacial and subglacial conditions, however, remain poorly understood due to insufficient direct measurements, which complicates the initialization and validation of thermal models. Here, we investigate the impact of using different geothermal heat flux (GHF) datasets and vertical velocity profiles on the thermal state of the Antarctic ice sheet, and compare our modeled temperatures to in situ measurements from 15 boreholes. We find that the  temperature profile is more sensitive to vertical velocity than GHF. The basal temperature of grounded ice and the amount of basal melting are influenced by both

selection of GHF and vertical velocity. More importantly, we find that the standard approach, which consists  of combining basal sliding speed and incompressibility to derive vertical velocities, provides reasonably good results in fast  flow regions (ice velocity $> 50$ m yr$^{-1}$), but performs poorly in  in slow flow regions (ice velocity $< 50$ m yr$^{-1}$). Furthermore, the modeled temperature profiles in ice streams, where bed geometry is generated using mass conservation approach, show better agreement with observed borehole temperatures, compared to kriging-based bed geometry.

- Tense of writing: When reading, I noticed that you switched between past and present tense a lot. I think the standard is to use the present tense. In the line comments, I tried to point out a few instances of when you used past tense, but I definitely did not catch all of the instances.

RC: Revised per the reviewer's comment, we have adjusted all statements to be in the present tense.

- Assumptions in the ice sheet model: Will the choice of a Budd sliding law impact the simulated thermal structure of the AIS? Same for the assumption that the effective pressure is equal only to the ice overburden pressure (meaning that you are assuming there is no subglacial water system at the ice-bed interface)? Several studies (e.g. Gustafson et al., 2022) have found a complex subglacial water system underlying the Siple Coast (where many of your borehole measurements are taken), which could certainly impact basal sliding (and thus vertical ice velocities). While I believe that an in-depth analysis of this is beyond the scope of this paper, it would be nice to see perhaps a figure or two in the supplement that show if your modeled thermal AIS states are sensitive to these two assumptions. I would also like to see limitations like this addressed in the discussion section as well.

RC: Thank you for this invaluable comment. This comment also provides an additional insightful perspective on how the basal conditions affect the ice velocity field, including vertical velocity. The general form of Budd type friction law can be written as general form of power type friction law:

$$\boldsymbol{\tau}_b = -CN^r \|\boldsymbol{v}_b\|^{s-1} \boldsymbol{v}_b \tag{1}$$

where $C$ is the friction coefficient, $\boldsymbol{v}_b$ is the basal velocity, $N$ is the overburden pressure, and $r = q/p$ and $s = 1/p$. In this study, we obtain the basal drag using $p = 1$ and $q = 1$. However, previous studies employ other types of friction law to investigate the future behavior of ice (Brondex et al., 2017; Yu et al., 2018). These alternative basal friction laws include:

- Weertman type friction law

$$\boldsymbol{\tau}_b = -C_w|\boldsymbol{u}_b|^{m-1}\boldsymbol{u}_b \tag{2}$$

where $C_w$ is the friction coeffient for Weertman friction law.

- Coulomb type friction law (Tsai et al., 2015; Brondex et al., 2017)

$$\boldsymbol{\tau}_b = \min(-C_w\boldsymbol{u}_b^m, fN) \tag{3}$$

where $f$ is the solid friction coefficient.

- Schoof type friction law (Schoof, 2005)

$$\boldsymbol{\tau}_b = -\frac{C_s|\boldsymbol{u}_b|^{m-1}\boldsymbol{u}_b}{1 + (\frac{C_s}{C_{\max}N})^{1/m}|\boldsymbol{u}_b|)^m} \tag{4}$$

where $C_s$ is the friction coefficient for Schoof type friction law, $C_{\max}$ is the maximum value of $|\boldsymbol{\tau}_b|/N$.

As the low basal velocity is estimated with the above friction laws, the current ice rigidity may not be not sufficient to reproduce the surface ice velocity. Aschwanden et al. (2012) incorporates an enhancement factor in effective ice viscosity to increase the vertical shear in SIA model as given by:

$$\mu = \frac{B}{2E\epsilon_e^{1/n-1}} \tag{5}$$

where $E$ is the enhancement factor. In slow flow regions, high rigidity leads to low vertical shear, hindering the flow of ice. Therefore, it is worth noting that conducting additional experiments with other types of friction laws and rigidity values is a viable option for reproducing the thermal regime of ice.

Additionally, we only consider that effective pressure is fully connected with ocean pressure. The changes in effective pressure due to subglacial hydrology system would affect the estimate basal velocity. While there are several tools available for modeling the subglacial hydrology system, such as GLaDS (Werder et al., 2013) and SHAKTI (Sommers et al., 2018), considering these models for calculating effective pressure is quite complex within the scope of our study.

Your invaluable suggestion expands our understanding of the thermal regime of ice. Therefore, we have included a discussion of your suggestion in the "Discussion" section.

Added part:

L280: In slow flow regions, we find that IVz-nosliding experiments show a reasonably good agreement with the observed borehole temperature profiles. However, the three-dimensional thermal model occasionally estimates convex temperature profiles, which are not consistent with the observations, such as KIS-1996-2 and KIS-2000-1,2 boreholes. Compared to other boreholes, the ice velocities at KIS and ER gradually decrease from upstream to downstream, and coincide with the presence of a basal ridge (Price et al., 2001; Ng and Conway, 2004) (see also Figure S2). Furthermore, the subglacial hydrology system at WAIS discharging to the Ross Ice Shelf has been explored using magnetotelluric, passive seismic data, and drilling borehole (Fisher et al., 2015; Priscu et al., 2021; Gustafson et al., 2022). The deceleration of tributaries at KIS and ER is attributed to water-piracy hypothesis (Alley et al., 1994) or removal of basal water contributing to the loss of lubrication (Tulaczyk et al., 2000; Bougamont et al., 2003). In model experiments, Bougamont et al. (2015) revealed changes in the tributaries at KIS and ER using a plastic till deformation friction law including simple subglacial hydrology model. In contrast, we employ the Budd type friction law and assume the effective pressure fully connected to ocean part, not including changes in the effective pressure. The variation in effective pressures also changed the basal ice velocity in Budd type friction law. In addition, a selection of other types of friction law, including Weertman (Weertman, 1974), Schoof (Schoof, 2005), and Coulomb (Tsai et al., 2015) types, also influences the initialization and future fate of ice (Brondex et al., 2017, 2019). Further investigation is required, such as the application of other types of friction laws or initialization with paleo spin-up, to better understand temperature profiles.

**Line Comments**

- L7: Can you be more specific when saying "vertical velocity plays a more important role in the temperature profile than GHF"? Do you mean that the temperature profiles are more sensitive to vertical velocity than GHF?

RC: Yes, this sentence aligns with the reviewer's point. To clarify the meaning of the text, we have made the following revisions.

Modified part:

L7: We find that the  temperature profile is more sensitive to vertical velocity than GHF.

- L8: ... which consists of combining ...

RC: Revised as reviewer's comment.

- L21: "Several important properties, such as ice elevation **and** surface ice velocity, ..." "... subglacial properties, such as ice temperature **and** geothermal heat flux, ..."

RC: Revised as reviewer's comment.

- L25: A little confusing wording, perhaps change to: In order to get reasonable estimates of these englacial and subglacial fields, inversion techniques are routinely employed (citations)."

RC: Revised as reviewer's comment.

Modified part:

L28: In order to get  reasonable estimates of these englacial and subglacial fields, inversion techniques are routinely employed to estimate basal friction and ice shelf rigidity (MacAyeal, 1993; Khazendar et al., 2007; Morlighem et al., 2010; Gillet-Chaulet, 2020).

- L28: "geothermal heat flux" should not be capitalized

RC: Revised as reviewer's comment.

- L29-30: " ... and ice dynamics (citations); **yet**, large uncertainties ..."

RC: Revised as reviewer's comment.

Modified part:

L32-34: ice dynamics (Pattyn et al., 2008; Seroussi et al., 2017; Smith-Johnsen et al., 2020b); yet, large uncertainties in spatial variation and magnitude of GHFs in Antarctica still remain.

- L36: " . . . better understanding of **subglacial** and englacial environments . . ."

RC: Revised as reviewer's comment.

- Methods: You use past tense here (e.g. "We used ISSM", "We used a 3D HO model", ...). I mentioned this above, but please try to switch this to present tense when possible.

RC: Revised as reviewer's comment.

- L76-78: Which surface ice velocity map did you use to perform the inversions?

RC: We used MEaSUREs version 2 ice velocity map (Rignot, 2017), so we have revised the manuscript as follow.

Modified part:

L82-83: To minimize misfit between modeled and observed ice velocities, the surface ice velocity of MEaSUREs version 2 (Rignot, 2017) is used. The ice rigidity under grounded ice was is estimated using the temperature-rigidity relation (Cuffey and Paterson, 2010).

- L102:106: In L102, you state that there are three different vertical velocity profiles, but in the experiment description, you only describe 2. Also, in L105, do you mean IVz-nosliding ignores the inferred basal sliding velocities?

RC: We apologize for the confusion. This is simply a typographical error. Indeed, we conducted two experiments and the latter part also aligns with what the reviewer mentioned. So, we have made the following revisions.

Modifie part:

L107-111: We computed compute the thermal state of the ice sheet using three two different vertical velocity profiles: 1) vertical velocity computed by solving for incompressibility while accounting for the inferred basal sliding (hereafter IVz), and 2) the equation of incompressibility of ice while not allowing basal sliding when surface ice velocities are below 10 m yr$^{-1}$ (hereafter IVz-nosliding). In other words, IVz IVz-nosliding ignores the inferred basal sliding velocities from the initial inversion and assumes that the bed is frozen when surface velocities are ¡ 10 m yr$^{-1}$.

- L115: It would be interesting to see a figure showing these four GHF data sets (with overlaid positions of borehole measurements) rather than only giving the AIS-means. I see you do this for the top row of figure 4, maybe reference that here and add borehole locations onto those maps if possible.

RC: The manuscript has been revised in accordance with the reviewer's comments. We have added borehole locations to the GHF field and have depicted it in the figure below. In the revised manuscript, this figure is now labeled as Figure 5 for reference.

[Figure]

Figure 1: Figure 5(in revised manuscript). Upper panels (a-d) are the geothermal heat flux distributions of each source. Middle panels (e-l) are the basal melting rate distributions, with the value at the bottom left indicating the total grounded ice melting volume for each experiment. The basal melting rate exceeding 50 mm yr$^{-1}$ is truncated. Lower Panels (m-q) are difference in basal melting rate between IVz-nosliding and IVz for each geothermal heat flux. A green cross dot on the geothermal heat flux map indicates the borehole location. The color map for difference in basal melting rates is from Crameri et al. (2020).

- L127: " ...between temperature measurements along the borehole profile and **triangular** mesh were not ..."

RC: Revised as reviewer's comment.

- L156: Add a space between sentences.

RC: Revised as reviewer's comment.

- L163: Remove double-comma

RC: Revised.

- L172: Change LVZ to LVz

RC: Revised.

- Table 3: This table is quite large and it is very difficult to comprehend results from it because there are so many numbers. I am wondering if there is a way you could add shading to the table to show greatest to least misfit (almost like a heatmap). Or you could add highlighting to the lowest misfit for each borehole (replacing the bold text, which does not stand out very much). Alternatively, I'm wondering if this data would be best visualized as a figure rather than a table?

RC: In accordance with the excellent suggestion from the reviewer, we have restructured Table 3 into a figure as shown below. In the revised manuscript, this figure has been designated as Figure 1 for reference.

[Figure]

Figure 2: Weighted absolute misfit between observed and modeled borehole temperatures according to each experiment. The absolute temperature misfit is truncated over 5°C. An asterisk in the AIS/WIS and BIS boreholes indicates the fast flow region.

- L181-185: Maybe I am confused, but in figure 2, it seems like the IVz-nosliding group captures the linear shape of the temperature profiles quite well (hence the good match between observations and the dotted lines for the first three profiles). Also, are we interested in the sign of R2, or just its absolute value of? Because while it is true that R2 is larger for IVz-nosliding than IVz, they both seem equally close to 0 when only considering the absolute value. Perhaps a description on how to interpret the R2 value in the methods section would help clarify this.

RC: We initially use the $R^2$ value as it typically indicates how well a linear line fits dispersed data. However, based on the reviewer's feedback and additional discussion, we have concluded that the $R^2$ value may not be a suitable metric for this study. Therefore, in the revised manuscript, we have replaced the use of the $R^2$ value with the correlation coefficient between the observed and modeled temperatures, as shown in Figure S1 of the revised manuscript (Figure 3).

[Figure]

Figure 3: Scatter plot showing the relationship between modeled and observed temperatures at each borehole depending on each experiment. Cross and triangle dot indicate IVz and IVz-nosliding group, respectively. A dashed solid line indicates modeled temperature equal to observed temperature.

• L229: Degree symbol is not a superscript in 5.1DegC

RC: Revised as reviewer's comment.

• L235-240: This finding about bed topography improving the performance of the thermal model is really great! I know it is not a main point of the paper, but I think it is super important and should be highlighted if possible (maybe in the discussion and/or conclusion sections?).

RC: Thank you for providing an opportunity to emphasize important findings. In response to the reviewer's comments, additional content has been incorporated into the discussion and conclusion sections as follows.

Added part:

L311: Thermal models have been used to reconstruct the thermal regime of ice and estimate the melting volume beneath grounded ice. Regarding the advection term in the thermal model, horizontal ice velocity is estimated with Higher Order or Full Stokes (FS) models, while the vertical velocity is recovered with the ice incompressibility. Under kriging-based geometry, the vertical velocity in fast flow region does not coincide with physical property. In contrast, state-of-the art bed geometry, such as BedMachine (Morlighem et al., 2017, 2020), is generated with the mass conservation, which of equation is based on ice incompressibility. We confirm that using the equation of ice incompressibility to reconstruct the ice vertical velocity provides a viable way of computing temperature profiles that exhibit good agreement with observations in Siple coast fast flow regions, such as the BIS. Given that the geometry of other fast flow regions, such as Thwaites Glacier, is generated using the mass conservation method (Morlighem et al., 2011, 2020), therefore, we expect that this study provides reliable temperature profiles. Note that the good agreement in modeled temperature at fast flow region, not only Siple coast fast flow region, does not guarantee the magnitude of basal melting volume because the basal melting volume at fast flow region is associated with the frictional heat. However, at slow flow region, the basal temperature is mainly affected by the GHF and the vertical advection, rather than the low frictional heat. It is worth noting that the basal melting volume would be reliable with IVz-nosliding.

• L243: Do you know why the vertical profile for Maule-IVz shows such high misfit for the AIS/WIS boreholes, whereas it always seemed fairly similar to the other IVz profiles?

RC: It is true that the modeled temperature profiles are sensitive to advection in ice, which is why the Maule experiment generally appears quite similar to most other IVz profiles.

To address the reviewer's question, upon conducting a more detailed analysis of the existing experimental results, we recognized an issue with the model producing abnormally cold values near the boundaries. In the process of improving this, we also realized that there is room for reducing discrepancies with observations, particularly in the AIS/WIS region, for the Maul-IVz-nosliding experiment.

After making these improvements, we performed additional sensitivity experiments to address the 2nd reviewer's suggestion to conduct additional experiments with different 2-m air temperature datasets (ERA5) and different flow region boundary for IVz-nosliding experiment with vel < 15, 18, 20 m/yr. As a result, while it is true that the modeled vertical temperature profiles in the Maule-IVz-nosliding experiment still exhibit larger discrepancies with observations in the AIS/WIS region, they have been improved to better match the observed profile shapes compared to the previous version (see Figure 2 in the revised manuscript).

Additionally, by considering the cumulative results of these additional experiments, we have concluded that the modeled temperature profiles are sensitive to the ice vertical velocity fields, and ice vertical velocity is also sensitive to ice rigidity.

- L246: By peripheral region, do you mean coastal regions? It might be helpful to use a more descriptive word here.

RC: The term "peripheral region" refers to the main ice trunk, so this sentence has been modified as follows.

Modified phrases:

L242-244: The mean basal temperature at the main ice trunk, where the ice  primarily discharges into the ocean, reaches the ice pressure melting point.

- L257: Perhaps here, it would be better to reference figure 3, where you show the basal temperature fields for each experiment. In figure 3, it is easy to tell that basal temperatures are warmer in IVz-nosliding compared to IVz; however, it is hard to distinguish differences in figure 4 e-l (see figure 4 comments below).

RC: As suggested by the reviewer, we have changed the reference figure from Figure 5 (4 in previous version) to Figure 4 (3 in previous version).

Line 251: All the experiments generally indicate that most of the regions experiencing basal melting are concentrated in fast flow regions, where basal frictional heat is significant and provides enough heat for the ice to reach the pressure melting point (Figure 4). Since IVz-nosliding displays lower vertical advection than that of IVz, the basal temperature of the IVz-nosliding group in slow flow regions is warmer than that of IVz (Figure 4c-j)

- L262-263: Hard to tell differences in grounded ice melt between GHF sources, see figure-4 comments.

RC: Following the reviewer's feedback, we have incorporated a figure illustrating the difference in basal melting rate between the IVz and IVz-nosliding experiments into Figure 1 (in this manuscript, Figure 5 in revised manuscript).

- L340: change "velocitiy" to "velocity"

RC: Revised as following reviewer's comment.

- Conclusions: I think it might be worth mentioning that varying spatial distributions of GHF did have a large impact on the spatial distribution of grounded ice melting rates across the AIS.

RC: We have incorporated the reviewer's mention of the effect of GHF into the conclusion.

Modified conclusions:

**Conclusions**
In this study, we used a three-dimensional thermo-mechanical model of Antarctica with different sources of GHF and vertical velocity fields to reproduce different thermal states of the Antarctic ice sheet, and we compared the results to 15 in situ measured

borehole temperature profiles in slow and fast flow regions. Comparing the modeled to measured borehole temperature profiles, we  confirm that the vertical ice  velocity based on the equation of incompressibility (IVz) is suitable for fast flow regions, such as BIS, where the bed geometry is constructed with using the mass conservation method, while an IVz that ignores basal sliding (IVz-nosliding) performs better in slow flow regions. Our results show that the vertical temperature profile  is more sensitive to the vertical velocity. In addition, the basal conditions, such as temperature and melting rate, are both sensitive to both GHF and the vertical velocity field. The total grounded ice melting  volume and basal temperature are proportional to the magnitude of the average GHF values for the same vertical velocity method. Finally, constraining the basal velocity to zero in slow  flow regions is a reasonable assumption and leads to a more realistic temperature profile.

- L343: Here, you say that the effects of different GHFs have little influence on the variance in basal temperature, but I would argue that figure-5 shows the opposite. Comparing figure 5c and 5d (the SR-IVz versus Maule-IVz basal temperature), the area of the ice sheet base that reaches the pressure melting point is much larger for the Maule-GHF than the SR-GHF (especially in East Antarctica). In fact, this difference in the hatched-white area is greater than that when comparing SR-IVz to SR-IVz-nosliding in this same figure, possibly showing that GHF has more of an impact on ice basal temperature than the vertical ice velocity. As it reads now, I think your conclusions underplay the importance of the GHF field in driving variance in ice basal temperatures.

RC: We have revised the conclusion in line with the reviewer's feedback, and the updated conclusion is included in the attached response to the query above.

- L353-355: Fix formatting here.

RC: Revised as reviewer's comment. We fix formatting the urls.

**Figure Comments:**

- Figure 2: It would be helpful to show visually which borehole measurements were taken in fast-flowing regions. Perhaps in these panels, you could add an asterisk or some identifier.

RC: Thank you for this comment. To enhance readability for readers, we have changed the placement of the AIS/WIS and BIS borehole results to the bottom row. Additionally, we have added blue and red boxes to emphasize and distinguish between slow flow and Siple coast fast flow regions, respectively.

[Figure]

Figure 4: Observed and modeled vertical temperature profiles from eight different experiments at 15 borehole locations. Blue and red boxes indicate slow flow and Siple coast fast flow regions, respectively. The bottom elevation at each borehole is set with considering the ice thickness, as listed in Table 1. An asterisk on borehole name indicates that the drilling reaches the bed rock. RR and Styx boreholes do not reach the bed rock.

- Figure 3: This figure is really fantastic!

RC: Thank you for your appreciation of this figure.

- Figure 4: Add locations of borehole measurements onto GHF maps if possible. Also, the colormap of the basal melting rate figures is very washed out on the positive side. Perhaps try limiting the colormap to 0.05 m/yr or using a log-scale (with using gray-shading for basal re-freezing since you cannot use log-scale for negative values) to better show regions of grounded ice melt. It could also be interesting to see difference maps in basal melt between respective GHF experiments (e.g. SR-IVz minus SR-IVz-nosliding) as an additional row. Also, it could be helpful to the reader if you include the AIS-integrated grounded ice basal mass balance value in each melt plot so that readers can get a feel for how quantitatively different each result is.

RC: Thank you very much for providing suggestions to enhance the readability of this Figure 5. Based on the reviewer's feedback, we have made the following improvements to the figure, as shown in Figure 6 in the revised manuscript.

[Figure]

Figure 5: (in previous manuscript) Figure 4

- Add locations of borehole measurements onto GHF maps if possible.

- RC: In accordance with the reviewer's suggestion, we have added borehole locations to each GHF field in the top row of Figure 5 in the revised manuscript.

- Also, the colormap of the basal melting rate figures is very washed out on the positive side. Perhaps try limiting the colormap to 0.05 m/yr or using a log-scale (with using gray-shading for basal re-freezing since you cannot use log-scale for negative values) to better show regions of grounded ice melt. It could also be interesting to see difference maps in basal melt between respective GHF experiments (e.g. SR-IVz minus SR-IVz-nosliding) as an additional row.

- RC: Thank you for this suggestion. To improve the readability of the basal melting rate for each experiment, we have changed the unit of basal melting rate from m $yr^{-1}$ to mm $yr^{-1}$, and set an upper limit 50 mm $yr^{-1}$. Due to the presence of negative values of basal melting rate, we could not use a logarithmic scale.

- Also, it could be helpful to the reader if you include the AIS-integrated grounded ice basal mass balance value in each melt plot so that readers can get a feel for how quantitatively different each result is.

- RC: Revised as reviewer's comment. We have added total grounded ice melting volume to the middle panels of grounded ice melting rate figures (Figure 5 in the revised manuscript).

[Figure]

Figure 6: (in revised manuscript) Figure 5. Upper panels (a-d) are the geothermal heat flux distributions of each source. Middle panels (e-l) are the basal melting rate distributions. The basal melting rate over 50 mm yr$^{-1}$ is truncated. Lower Panels (m-q) are difference of basal melting rate between IVz-nosliding and IVz for each geothermal heat flux. Colormap for difference in basal melting rate is from Crameri et al. (2020).

- Figure S2: There are very sharp transitions in B in the IVz-nosliding panel (see annotated figure below), especially along interior sectors of the EAIS. Does this occur because these are the locations where basal sliding cuts off to 10 m/yr? It might be worth addressing this in the manuscript.

RC: As you mention, the sharp transition in B (ice rigidity) is related to where basal sliding cuts off to 10 m/yr. We mention this sharp transition zone and some futher works on this problem.

L340:Furthermore, the adoption of no-sliding in specific regions results in a sharp transition zone in ice rigidity, B. This occurs because the basal velocity near the transition zone does not smoothly changed from no-sliding to sliding (Figure S4). Therefore, additional work is required to address and resolve the transition between no-sliding and sliding.


September 24, 2023

We would like thank the anonymous reviewer for their insightful and valuable comments, which have helped us to improve our manuscript. We address the remarks below point by point. To clarify the reviewer's comment and its reply, we have used red for the reviewer comment and black for the replying comment (RC).

**1 Reviewer #2 (Anonymous Reviewer)**

**Summary:**

The manuscript presents simulations of Antarctica to investigate the sensitivity of the modeled thermal state to the boundary conditions and inversion method using the three-dimensional thermomechanical ice sheet model, ISSM. They focus on the influences caused by differences in existing geothermal heat flux maps and the effect of differences in the ice vertical velocity. Both GHF and vertical velocity are poorly constrained in models but are known to affect the thermal state. The authors provide a new set of model simulations with different combinations of GHF maps and vertical velocity parameterizations to generate 3D temperature fields. By comparing their modeled temperature fields to existing borehole temperature profiles, the authors conclude that vertical velocity has a greater influence on the thermal state than GHF. This new contribution is very compelling, since it implies that vertical velocity is critical to constrain in ice sheet models in order to accurately model the thermal state. However, the authors miss the opportunity for some additional analysis and discussion which will further strengthen their findings and narrative.

**Major Issues**

The main findings are ice-sheet scale conclusions about the effect of boundary conditions and model initialization on the thermal state, while the boreholes that are analyzed to make these conclusions are largely from the Siple coast, which has a unique thermal configuration (Englehardt 2004, Bougamont et al., 2015, Ng and Conway 2004, etc). The stagnation of Siple Coast ice streams (i.e. Kamb > 150 years ago) exhibits an interesting thermal regime today that must contain memory of the past slow down. However, this would not be the case for most other parts of Antarctica. I think the paper should be strengthened in two ways.

- The authors should add more discussion on the Siple Coast model/observation discrepancies since it is very interesting. I think there is a missed opportunity to elaborate on the convex vs. concave temperature profiles for models vs. observations for the KIS and ER profiles (Fig. 2). I wonder if the difference in shape is an indication that the model's lack of the long-term thermal state memory really matters for this region. I know this shortcoming is hinted at in the last paragraph of the discussion, but I think the authors miss the chance to add interesting discussion about what the discrepancies in model vs observed temperature profiles are telling us about the thermal regime. I'm not expecting new results, but I would like to see some speculation about the effects of boundary conditions vs. initialization approach (inversion with present day conditions, paleo spin-up, thermal steady state approximation, etc) in the discussion.

RC: We appreciate your invaluable comment. Your suggested references, particularly the Kamb Ice Stream (KIS or UpC reference), have been immensely helpful in enhancing our understanding of the ice dynamics processes. Previous studies (Bougamont et al., 2015) successfully replicated the concave shape observed in KIS and ER borehole temperatures, opposite to the typical shape of borehole temperature, by adopting a plastic basal boundary condition for higher order model. A plastic boundary condition is given by

$$\tau_b = -a \ \exp(-be) \frac{\mathbf{u}_b}{\|\mathbf{u}\|}, \tag{1}$$

where $a$ and $b$ are two positive empirical constants, $e$ is void ratio, and $\mathbf{u}_b = (u, v)$ is basal ice velocity, $\|\mathbf{u}\| = \sqrt{u^2 + v^2 + \gamma^2}$, and regularization term with $0 < \gamma \ll (u, v)$. We believe that plastic till deformation approach is a crucial in addressing the issues observed in KIS and ER boreholes.

Consequently, we have incorporated this discussion into the manuscript as follows:

Line 286-294: The deceleration of tributaries at KIS and ER is attributed to water-piracy hypothesis (Alley et al., 1994) or removal of basal water contributing to the loss of lubrication (Tulaczyk et al., 2000; Bougamont et al., 2003). In model experiments, Bougamont et al. (2015) revealed changes in the tributaries at KIS and ER using a plastic till deformation friction law including simple subglacial hydrology model. In contrast, we employ the Budd type friction law and assume the effective pressure fully connected to ocean part, not including changes in the effective pressure. The variation in effective pressures also changed the basal ice velocity in Budd type friction law. In addition, a selection of other types of friction law, including Weertman (Weertman, 1974), Schoof (Schoof, 2005), and Coulomb (Tsai et al., 2015) types, also influences the initialization and future fate of ice (Brondex et al., 2017, 2019). Further investigation is required, such as the application of other types of friction laws or initialization with paleo spin-up, to better understand temperature profiles.

- If the focus of the paper is on the broad scale effects of GHF vs vertical velocity on thermal regime, then there are more borehole temperature profiles, which should be added to the analysis. There are more from East Antarctica such as Dome C, Lake Vostok, Talos Dome, South Pole. There are also more borehole temperature profiles in other parts of West Antarctica such as Kohnen and Byrd (maybe some others I am missing?).

RC: We appreciate the insightful comment from the reviewer and have made additional efforts to address these suggestions. In response to your comment, we have used polynomial functions from Talalay et al. (2020), which provide polynomial functions describing the relationship between temperature and depth from the surface (Table 1). As we have already obtained and utilized data from Dome Fuji and WAIS Divide, we have focused on utilizing borehole temperatures from Byrd, Vostok, Dome C, and Kohnen from Talalay et al. (2020).

Table 1: The polynomial approximation depicting the relationship between borehole temperature $T(°C)$ and vertical depth $z$ (m). The polynomial function is listed on Table 3 in Talalay et al. (2020)

| Drill sites | Polynomial |
| --- | --- |
| Byrd | $T = -28.343 + 0.8367 \times 10^{-3}z - 6.7651 \times 10^{-6}z^2 + 6.1339 \times 10^{-9}z^3$ |
| Vostok | $T = -56.034 + 2.9889 \times 10^{-3}z + 3.888 \times 10^{-6}z^2 + 0.2419 \times 10^{-9}z^3$ |
| Dome C | $T = -54.316 + 5.2978 \times 10^{-3}z + 4.4141 \times 10^{-6}z^2 - 0.368 \times 10^{-9}z^3$ |
| Kohnen | $T = -44.428 + 1.7384 \times 10^{-3}z + 4.4124 \times 10^{-6}z^2 + 0.184 \times 10^{-9}z^3$ |

Figure 1 displays modeled and observed borehole temperatures at Byrd, Vostok, and Kohnen. Note that we do not adjust the surface temperature to match the observed temperature, which is why the surface temperature from ERA-Interim displays a large offset from top of borehole temperature. For Byrd and Vostok, considering the drilling borehole depth, the observed temperature derived from polynomial approximation displays notably high temperatures below 1700 m and 3000 m. Consequently, we truncate observed temperature below a specific depth (Figure 1). Despite the surface temperature mismatch, the modeled temperature at Vostok, Dome C, and Kohnen somehow captures the linear shape of observed borehole temperature (Figure 1).

We made efforts to collect additional borehole logging profiles, including RABID project, Whillans, WACSWAIN project described below.

1. RABID project

    – Hot water drilling at Rutford Ice Stream, West Antarctica (Smith, 2005)
    – Measure borehole temperature using thermistor string, which of length is about 300 m (Smith, 2005, p. 20)

2. Whillans Ice Stream Subglacial Access Research Drilling (WISSARD) project

    – How water drilling at Subglacial Lake Whillans.
    – Measure borehole temperature distributed temperature sensing (DTS) (Fisher et al., 2015).

3. WACSWAIN project (Mulvaney et al., 2021)

    – Drilling borehole at Skytrain Ice Rise (see Figure 4 at Mulvaney et al., 2021)

However, we were unable to obtain actual vertical temperature profile data. There might be other drilling projects that we are unaware of.

While the validation of our research would significantly benefit from the inclusion of additional borehole temperature profiles, unfortunately, we were unable to obtain any additional borehole temperature profiles. Actually we invested a significant amount of time in collecting borehole temperature profiles; however, accessing borehole temperatures is often challenging.

[Figure]

Figure 1: Modeled and observed borehole temperatures at Byrd, Vostok, Dome C, and Kohnen. The observed borehole temperatures are derived from the polynomial approximation from (Talalay et al., 2020)(also see Table 1). A red vertical line indicates the observed surface temperature. For Byrd and Vostok, temperature data is limited to depth of 1700 m and 3000 m because unusual temperature is calculated below these depths with polynomial approximation.

[Figure]

Figure 2: Measuring temperature using distributed temperature sensing (DTS). This graphic is from Figure 3 at Fisher et al. (2015).

[Figure]

Figure 3: Observed and modeled borehole temperature profiles at Skytrain Ice Rise (Figure 12 from Mulvaney et al., 2021).

Regarding the naming of "fast flow" and "slow flow" regions, somewhere early on in the manuscript, it should say that it isn't possible to drill into most really fast flowing regions because deformation etc. prevent drilling. In the paper I would say "fast flow" only defines a unique subset of ice streams (WIS and BIS) where the flow regime supports drilling, so there are borehole temperature profiles for only those regions. Because of this, the "fast flow" conclusions may not apply for other parts of Antarctica. I might even recommend renaming "fast flow" to "Siple coast fast flow" or something like that throughout the manuscript to clarify this point.

RC: Thank you for your comment. As suggested by the reviewer, we have modified notation of "fast flow" to "Siple coast fast flow" in revised manuscript.

Modified phrases:

> Line 163-165: Note that AIS/WIS-1991-1, AIS/WIS-1995-4,7, and BIS-1998-4,5 , are located in regions with comparatively high ice velocity compared to other boreholes and have concave temperature profiles. To clearly define this specific fast flow region, we refer to AIS/WIS and BIS as Siple coast fast flow region.

There are known difference in 2m air temperature amongst reanalysis products such as ERA-interim, ERA5, RACMO, MERRA, MAR. Why do the authors choose ERA-Interim? The author's test the effect of the basal boundary condition by changing GHF maps, while the surface boundary condition is never tested. It would be helpful to see additional simulations using different 2m air temperature products to see its effect on the vertical temperature profiles, even if this effect is less significant.

RC: We appreciate you pointing out the aspect we overlooked. We have conducted additional analysis to address your question.

Figure 4 displays the climatological mean 2-m air temperature of ERA-Interim (Dee et al., 2011) (hereafter ERAI), ERA5 (Hersbach et al., 2023), RACMO2.3p2 forced with ERA5 (hereafter RACMO23p2-ERA5) (van Wessem et al., 2023), and MERRA2 (Global Modeling and Assimilation Office (GMAO), 2015). The 2-m air temperature data from ERAI shows less variability than the others, and its smoothed nature might compromise accuracy (Figure 4). While ERA5, RACMO23p2-ERA5, and MERRA2 are relatively recent datasets, they exhibit some differences in the climatological mean 2-m air temperatures when compared to the observed surface temperatures at each borehole (Figure 5). Consequently, it is apparent that surface ice temperature corrections are also necessary for ERA5, MERRA2, RACMO23p2-ERA5. Moreover, through a simple exponential decay correction considering differences between the reanalyses and observations (Figure 4 ), it's evident that the corrected ERAI and ERA5 data results closely align with the observed values,

surpassing their initial counterparts. Therefore, we have selected ERA5 from these reanalyses and conducted additional experiments following the same experimental design.

The additional experiments conducted with 2-m air temperature from ERA5 display no significant differences compared to experiments with ERAI (Figure 6). However, in case of SD, RR, and AIS/WIS (only applicable in the case of nosliding), there are slight discrepancies in surface temperature leading to shifts in the modeled temperature profiles when using ERA5. In fact, ERA5 corrections at these specific locations have improved results, making them more similar to observaiton.

An important conclusion drawn from this series of additional experiments is that surface ice temperature, or the accuracy of its correction, significantly influences the simulated ice temperature profiles within the model, and this information has been incorporated into the discussion section.

> **Discussion**
> Line 350: The surface temperature of ice would be one of factors to consider the boundary condition of thermal model. While ERA5 (Hersbach et al., 2023), RACMO2.3p2 forced with ERA5 (van Wessem et al., 2023), and MERRA2 (Global Modeling and Assimilation Office (GMAO), 2015) are the recent reanalysis datasets, they display some discrepancies between the climatological mean 2-m air temperature (1980-2018) and observed surface temperature at each borehole (Figure S6). For the comparison with different version of ECMWF (European Centre for Medium-Range Weather Forecasts) reanalysis data, we perform experiments using the same manner, utilizing 2-m air temperature from ERA5. These results display no significant differences compared to experiments using ERA-Interim (Figure S7). However, in case of SD, RR, and AIS/WIS (only for the IVz-nosliding case), they display slight discrepancies in surface temperature leading to shifts in the modeled temperature profiles. In fact, the improvement in surface temperature and the accurate correction would bring the modeled temperatures into closer agreement with observations.

[Figure]

Figure 4: Climatological mean 2-m air temperature for (a) ERA-Interim, (b) ERA5, (c) RACMO23p2-ERA5, and (d) MERRA2 during 1979-2018.

Table 2: Velocity misfit between modeled and observed surface ice velocity and total grounded ice melting volume for IVz and IVz-nosliding forced with ERA5.

| GHF | Velocity misfit (m yr$^{-1}$) | | Total grounded ice melting volume (Gt yr$^{-1}$) | |
|---|---|---|---|---|
| | IVz | IVz-nosliding | IVz | IVz-noslding |
| Shapiro | 13.01 | 18.96 | 21.29 | 24.18 |
| Fox | 13.48 | 19.28 | 25.22 | 28.54 |
| An | 12.95 | 19.93 | 16.87 | 19.39 |
| Martos | 13.01 | 18.01 | 25.32 | 28.18 |

[Figure]

Figure 5: Difference between the climatological mean 2-m air temperature and observed surface temperature at each borehole. The climatological means for ERAI, ERA5, RACMO23p2-ERA5, and MERRA2 are from the period 1979-2018. Borehole names highlighted in red indicate where surface ice temperature is corrected using exponential decay. $T_{\mathrm{corr}}$ indicates the difference between the corrected and observed temperatures for ERAI and ERA5.

[Figure]

Figure 6: Differences between observed and the mean modeled vertical temperature profiles depend on each experiment group. A black vertical dashed line indicates where the misfit is zero.

[Figure]

Figure 7: (a,c) Temperature difference between corrected and original. (b,d) Number of over-lapping region where a interference radius is set to 10 km. Magenta triangle indicates borehole locations used in this study. Region at (c,d) is zoomed into the WAIS discharging to Ross Ice Shelf, where most boreholes are located.

Also on the topic of surface temperature, in Fig. 2, it looks like the surface temperatures between the ISSM simulations and observation are not a great match for some boreholes (e.g. KIS, WIS, ER, Bruce, UC, ER, SD). I see in the text it says that the model surface temperatures are adjusted using an exponential decay function to better match the observations so I would like to know why there is this miss match. Would a different 2m air temperature map provide a better match to observations needing less correction (see comment above)?

RC: The correction for surface temperature is effective at Fuji Dome, Styx Glacier, WAIS Divide, and Law Dome (Figure 5). As the boreholes at UC, ER, SD, and AIS are in close proximity (Figure 7c,d), each corrected temperature can influence the others. In the exponential decay correction, a 10 km radius results in approximately 80% differences ($\exp(-10/50) \approx 0.81$), and impacts nearby boreholes. Therefore, instead of applying the correction to all boreholes, we have selected specific ones (highlighted in red in Figure 5).

How was 10m/yr threshold for surface velocity chosen for the IVz-nosliding experiment? Are the results sensitive to nudging this threshold? I am not necessarily asking for more model simulations here, but I would like to better understand the choice and its likely effect on the results.

RC: We would like to clarify that we based our experiments on an ice velocity threshold of approximately 10 m/yr, as most boreholes in the slow flow regions observed ice velocities below this value.

In response to the reviewer's comments, we have conducted additional experiments in the IVz-nosliding group, where we set the nosliding slow flow region velocities to 15, 18, and 20 m/yr, respectively. To maintain clarity in naming each experiment, we followed a consistent format. For instance, we have denoted IVz-nosliding with velocities less than 15 m/yr as IVz-nosliding15, and similarly for the other experiments.

Figure 8 displays discrepancies between the mean modeled and observed temperatures. In comparison to IVz-nosliding10, the additional experiments generally exhibit similar trends in most regions, albeit not replicating its values accurately (Figure 10). This can be attributed to the fact that the absence of sliding in specific regions affects the stress equilibrium of the ice and changes in ice velocity also affects the thermal regime. The new experiments slightly improve results in Law Dome, ER, and KIS regions, prompting a need for deeper investigation into the causes. Nonetheless, the more significant finding is that the analysis of additional experiments indicates that there is little change in the total grounded ice volume as the nosliding slow flow regions expand within IVz-nosliding (Figure 10).

These additional experiment results confirm that changing the criteria for nosliding slow flow regions does not significantly impact the key findings. In addition, these experiments reveals that higher nosliding slow flow region boundary results in higher misfit in initialized surface ice velocity. Therefore, using the existing results with a threshold of 10 m/yr to minimize misfit is a valid approach and does not pose significant issues, especially given the minimal differences observed.

[Figure]

Figure 8: Differences between observed and the mean modeled vertical temperature profiles depend on velocity boundary for IVz-nosliding. Solid and dash line indicate original and additional experiments, respectively. Vertical dash line indicates zero misfit line.

[Figure]

Figure 9: Velocity misfit for (a) whole domain and (b) region where ice velocity is over 50 m yr$^{-1}$. Horizontal dot line indicates mean velocity misfit of IVz group. Triangle marker indicates original IVz-nosliding experiment.

[Figure]

Figure 10: Total grounded ice melting volume depending on IVz-nosliding experiment with different slow flow region boundary. Horizontal line indicates IVz experiment.

**Minor Issues**

In the abstract, fast flowing has a velocity threshold definition but not slow flowing. This should also be provided. Is it slower than 50 m/yr or something else? The reasoning behind these choices should also be explained somewhere near the beginning of the manuscript. For example, Dawson et al., 2022 uses 100 m/yr to define fast flowing regions. Are these thresholds a result of the velocities seen at the boreholes and some natural separation in the velocities/profiles?

RC: Thank you for pointing out the aspect we overlooked. To define the fast vs. slow flow region, we relied on the critera outlined in Seroussi et al. (2011). According to Seroussi et al. (2011), the regions with ice velocity $> 50$ m yr$^{-1}$ displays that a ratio between depth-averaged ice velocity and surface ice velocity ($=\|\boldsymbol{u}\|/\|\boldsymbol{u}_s\|$) is approximately 99 % for Higher-Order (HO) model. Therefore, we designated regions with ice velocity more than 50 m/yr as fast flow region, where sliding dominates over internal deformation. In abstract, in accordance with your suggestion, we have added the definition that we consider areas with ice velocities below 50m/yr as slow flow regions.

It would be useful to see observed surface velocities at the boreholes reported in Table 1 so that the reader could see what borehole sites are within the model prescribed no sliding regions (as well as the fast and slow flow groupings). It's hard to get this information from Fig. 1 right now... perhaps if a 10m/yr contour was drawn on the map then that could also work.

RC: Revised as reviewer's comment. We have added white dot contour line indicating where ice velocity is 10 m/yr (Figure 11, see also Figure 1 in revised manuscript).

[Figure]

Figure 11: (a) Borehole locations with temperature measurements overlaid over ice velocity (Rignot, 2017). The black dashed box shows the location of (b). The black solid box in (a) indicates each basin from Jourdain et al. (2020), and each number indicates each basin number. We use different symbols for each borehole based on the shape of their temperature profile (triangle and cross red dots indicate concave and linear profiles, respectively). The gray contours indicate surface elevations, with dash lines for every 500 m and solid lines for every 1000 m. The white dot contours indicate regions where ice velocity is 10 m yr$^{-1}$. (b) Enlargement of borehole locations at West Antarctica overlain over the ice velocity. The borehole names are abbreviated: WIS, Whillans Ice Stream; BIS, Bindschadler Ice Stream; ER, Engelhardt Ridge; KIS, Kamb Ice Stream; RR, Raymond Ridge; UC, Unicorn; AIS, Alley Ice Stream; SD, Siple Dome.

The organization of the subplots in Fig. 2 is somewhat confusing to me. I think the first row are the "linear" borehole profiles and then the rest are the more "concave" profiles. It also took me a while to see that the bottom row of profiles are the ones from the fast flowing regions. I think it would be helpful to see the profiles boxed into a slow flowing group (where IVZ-nosliding fits the observations better) and a fast flow group (where IVZ fits better), like the subtle separation in Table 1.

RC: Revised as reviewer's comment. To make it more reader-friendly, we have added blue and red boxes indicating slow and Siple coast fast flow regions, respectively.

[Figure]

Figure 12: Observed and modeled vertical temperature profiles from eight different experiments at 15 borehole locations. Blue and red boxes indicate slow flow and Siple coast fast flow regions, respectively. The bottom elevation at each borehole is set with considering the ice thickness, as listed in Table 1. An asterisk on borehole name indicates that the drilling reaches the bed rock.

In Fig. 2, I would also find it helpful to see the boreholes that go all the way to the bed somehow indicated in Fig. 2.

RC: The elevation of each borehole profile is limited with considering ice thickness, as listed in Table 1. See also above Figure 12.

Line 69: HO should be defined as higher order, with appropriate citation given.

RC: Revised as reviewer's comment. Add Pattyn (2003) for High-Order model (HO) in the revised manuscript, L56-58.

Modified part:

> L57-58: Due to scarcities of internal ice velocity measurements, three-dimensional mechanical models, such as Higher-Order (HO; Pattyn, 2003) and Full Stokes (FS), are used to estimate internal ice velocities.

Line 79: Be clearer about what the temperature rigidity relation is (e.g. give page # in Cuffey).

RC: Revised as reviewer's comment.

Modified part:

> Line 83-84: The ice rigidity under grounded ice is estimated using the temperature-rigidity relation (Cuffey and Paterson, 2010, pp. 72–77).

Line 99: Give version of BedMachine

RC: Revised. We used BedMachine version 1, and it is clearly stated in the revised manuscript.

Modified part:

> Line 103-104: The bed geometry is from BedMachine version 1 (Morlighem et al., 2020),

For Fig. 4, I could see on the colorbar writing GHF instead of G to be more consistent with the text.

RC: Revised. We have changed colorbar title for GHFs with "GHF".

[Figure]

Figure 13: Figure 5 in revised manuscript.

Paragraph starting on line 292: I am confused what is being reported here. I think you mean total melt water volume rather than melting rates. Melting rates should be reported in mm/yr or m/yr (such as the author's Fig. 4 and Pattyn, Jouquin, Llubes) while total melt water volume is Gt/yr. This paragraph should be clarified what measure is being discussed. It would also be helpful if the authors state what their values are rather than just saying they are lower than Pattyn and higher than Llubes.

RC: Thank you for the insightful and constructive suggestion. To avoid confusion between "total grounded ice volume" and "melting rate", we change the notation of "total grounded ice melting rate" to "total grounded ice melting volume" in Table 4 and throughout the manuscript.

On line 308-309, elaborate more on the mass conservation $->$ melting rates $->$ understanding subglacial hydrology comment. I'm not sure I understand what this sentence is trying to say. I think the paragraph could use some rewriting to clarify the point.

RC: Revised as reviewer's comment.

Modified discussion:

> Line 311-324: Thermal models have been used to reconstruct the thermal regime of ice and estimate the melting volume beneath grounded ice. Regarding the advection term in the thermal model, horizontal ice velocity is estimated with Higher Order or Full Stokes (FS) models, while the vertical velocity is recovered with the ice incompressibility. Under kriging-based geometry, the vertical velocity in fast flow region does not coincide with physical property. In contrast, state-of-the art bed geometry, such as BedMachine (Morlighem et al., 2017, 2020), is generated with the mass conservation, which of equation is based on ice incompressibility. We confirm that using the equation of ice incompressibility to reconstruct the ice vertical velocity provides a viable way of computing temperature profiles that exhibit good agreement with observations in Siple coast fast flow regions, such as the BIS. Given that the geometry of other fast flow regions, such as Thwaites Glacier, is generated using the mass conservation method (Morlighem et al., 2011, 2020), therefore, we expect that this study provides a method to generate reliable temperature profiles. Note that the good agreement in modeled temperature at fast flow region, not only Siple coast fast flow region, does not guarantee the magnitude of basal melting volume because the basal melting volume at fast flow region is associated with the frictional heat. However, at slow flow region, the basal temperature is mainly affected by the GHF and the vertical advection, rather than the low frictional heat. Therefore, it is noteworthy that the basal melting rate produced using IVz-nosliding in slow flow regions would be reliable.

Regarding the data availability statement, I believe that this paper would have broader impact and community interest, myself included, if the ISSM thermal model results from this analysis were made available as part of this study. I recommend providing a link to download the gridded temperature fields or simply providing the ISSM outputs for each run. This would enable further comparisons and validation of thermal modeling efforts.

RC: Thank you for the valuable suggestion. In accordance with the reviewer's recommendation, we have shared the gridded basal temperature fields via KDPC (Korea Polar Data Center) with DOI in revised manuscript

Inconsistent with the use of Gt vs. Gton throughout the manuscript.

RC: Revised. We have used "Gt" as default unit in revised manuscript.

Writing style in general is mixing past and present tense, which should be resolved.

RC: Revised. We have modified to the present tense for all statements.

**Technical corrections**

- Typo on line 43: incompressbility

RC: Revised.

Line 47-48: The vertical velocities used in one-dimensional thermal model are generally recovered through the equation of incompressibility, assuming a stationary bed ...

- Mistake on line 107: "three" $-$ > "two"

RC: Revised.

Line 106: the thermal state of the ice sheet using  two different vertical velocity profiles:

- Typo on line 114: exptrapolate

RC: Revised.

Line 119: model to  extrapolate

- Typo on line 142: extra space after Y?

RC: The weighted correlation factor is removed in revised manuscript. As Dr. Tyler Pelle (other reviewer) commented this part and we have discussed and replaced the use of the $R^2$ value with the correlation coefficient between the observed and modeled temperatures, as shown in Figure S1 of the revised manuscript (Figure 14)

[Figure]

Figure 14: Scatter plot showing the relationship between modeled and observed temperatures at each borehole depending on each experiment. Cross and triangle dot indicate IVz and IVz-nosliding group, respectively. A dashed solid line indicates modeled temperature equal to observed temperature.

- Typo on line 156: Missing a space ("datasets.Table")

RC: Revised.

- Typo on line 163: two commas

RC: Revised.

Line 159: toward the bed dominates, while the other group has more linear shap

- Typo on line 226: indicates ← indicate

RC: Revised.

> Line 222: IVz-nosliding group; these values indicate high vertical advection toward the bottom.

Typo on line 306: delete "a" and "goods" → "good"

RC: Revised.

> Line 317: computing  temperature profiles that exhibit  good agreement with observations in Siple coast fast flow regions, such as the BIS

Typo on line 340: velocitiy

RC: Revised.

> Line 368: we confirm that the vertical ice velocity based on the equation of incompressibility (IVz) is suitable for fast flow regions, such

[revised manuscript text omitted]

---

## Author Response (AR2)

**Impact of boundary conditions on the modeled thermal regime of the Antarctic ice sheet**
**– Response to Editor's Review –**

We would like to thank Dr. Benjamin Smith for his insightful and valuable comments, which help us to improve our manuscript. We address his remarks below point by point. The editor's comments are shown in red, and our replies are in black (RC).

TThe KIS temperature profiles are likely from iche new version of the manuscript does a nice job of responding to the referee's concerns with the first version. There's one technical point (about KIS, below) that I'd like to see addressed throughout the study, and I have some edits I'd like to see to improve the clarity of the writing. In general, the manuscript could use another round of editing with an eye towards removing redundant phrases, and towards improving paragraph structure so that the topic sentence of each paragraph gives a clear idea of the ideas that are to be discussed in that paragraph.

The KIS temperature profiles are likely from ice that as moving quickly before the ice stream's stagnation around 1860 CE. As a result, the deeper ice likely came from places where the surface temperature was significantly lower that it is now, and the temperature profiles reflect strain regimes characteristic of basal sliding. This history should be taken into account in evaluating the models, which cannot be expected to know about the wayward behavior of KIS.

☑ Lines 124-132: This paragraph as written is hard to follow (too many specifics to address as lists within sentences). Consider conveying this information in a table.

Rather than discussing every borehole in the main text, we now refer to Table 1. The changed sentence is as follows:

Line 124: To validate the thermal models, we compile all available 15 borehole temperature profiles listed in Table 1. The 10 boreholes in the West Antarctic Ice Sheet region are drilled at Whillans Ice Stream (WIS), Bindschadler Ice Stream (BIS), Engelhardt Ridge (ER), Kamb Ice Stream (KIS), Raymond Ridge (RR), Unicorn (UC), Alley Ice Stream (AIS), and Siple Dome (SD) (Engelhardt, 2004a) (Figure. 1b).

☑ Line 146: Confusing: rewrite as "…except for at Dome Fuji and Law Dome, for which few thickness measurements were available."

Done

☑ Line 154: Too many significant figures in the misfit values. 12.5 and 19.5 are more than enough.

Done

☑ Line 155: "relatively lower" is redundant

This sentence is changed as follows;

"Line 152: The standard deviation in ice velocity misfit is 0.09 m yr$^{-1}$ for the IVz, and 0.35 m yr$^{-1}$ for the IVz-nosliding group."

☑ Table 2: too many significant figures in velocity misfits.

We updated the table as follows:

| GHF | Vertical velocity | |
|---|---|---|
| | IVz | IVz-nosliding |
| SR | SR-IVz | SR-IVz-nosliding |
| | (12.4 m yr$^{-1}$) | (19.9 m yr$^{-1}$) |
| Maule | Maule-IVz | Maule-IVz-nosliding |
| | (12.5 m yr$^{-1}$) | (19.1 m yr$^{-1}$) |
| An | An-IVz | An-IVz-nosliding |
| | (12.6 m yr$^{-1}$) | (18.6 m yr$^{-1}$) |
| Martos | Martos-IVz | Martos-IVz-nosliding |
| | (12.3 m yr$^{-1}$) | (19.7 m yr$^{-1}$) |

☑ 165-75: Consider the above point about KIS in relation to this material.

We appreciate the editor's comment. It is not easy to determine the exact period when the stagnation of the KIS region started. According to Joughin and Tulaczyk (2002), the ice has been stagnant there for about 150 years. We therefore assume that the start of stagnation year was around 1850 CE (150 years ago from 2002). We have modified the text as follows:

> "Line 173: However, none of the experiments successfully reproduce the temperature profiles at KIS boreholes, where the ice has been stagnant since around 1850 CE (Alley et al., 1994; Joughin and Tulaczyk, 2002). This history cannot be captured by our thermal steady-state assumption. A more detailed description of misfit values for each borehole can be found in the next section."

☑ 177: "Let's focus" is too informal.

This sentence is changed as follows:

> "Line 177: First, we focus on the three borehole profiles: SD, RR, and Dome Fuji."

☑ Line 180: It looks to me like the nosliding groups do quite well for Fuji and SD. Is this a typo?

Thank you for catching the typo. We fixed the typo as follows:

> "Line 180: For these boreholes, the IVz group does not capture the linear shape of the temperature profiles."

☑ Line 181 "which of value" –possible typo?

Revise the typo as follows:

"Line 181: The IVz-nosliding group at these boreholes has a misfit value within 2$^{\circ}$C, which is lower than that of the  IVz group (Figure 3)."

☑ Line 186: "The basal modeled temperature at An is the lowest…" – should be "for An"

Done.

☑ 190: "The borehole of Styx glacier is a shallow ice core". – Redundant. Please rewrite.

This sentence is revised as follows:

"Line 190: At the borehole of Styx Glacier, both IVz and IVz-nosliding groups display similar average misfit values of ~0.64°C and ~0.40°C, which show good agreement with the observed temperature profiles. The drilling depth of Styx Glacier is about 210.5 m (Yang et al., 2018), and the ice thickness measured with ground penetrating radar survey is about 550 m (Hur, 2013). While We cannot definitively confirm the basal condition from observations, the thermal model results suggest that none of the experiments reach the melting point."

☑ 194: UC was likely an area of stagnant ice in the middle of the old ice stream.

We added some information that UC boreholes are located at stagnant ice, and the sentence ice changed as follows:

"Line 195: The UC boreholes are located in an area of stagnant ice and have a relatively high basal temperature gradient compared to the other adjacent boreholes, such as AIS/WIS boreholes (Engelhardt, 2004b)."

☑ 225: refer to table 3, not figure 3

The temperature misfit value is listed in Figure 3, therefore, we think that this is the right reference.

☑ 227: Don't need "from MEASURES version 2" (the Rignot citation is enough).

This sentence is changed to

"Line 219: The observed ice velocity at Bruce Plateau is 49 m yr$^{-1}$ according to Rignot (2017),~"

☑ 228-229: Please explain why there is any difference at all between the sliding and nosliding models here (since both are sliding)

Although both the IVz and IVz-nosliding experiments allow for sliding in fast flow regions, the mean modeled ice velocity of the IVz-nosliding group in the AIS/WIS region is generally slower than the one of the IVz group (Figure 1). However, in the BIS region, it seems that the modeled ice velocity for IVz and IVz-nosliding are not significantly different, both consistent with observed ice velocity. The difference between these two cases is the width of the ice stream. When ice streams are narrow, the modeled speed will be sensitive to the no-sliding condition that is imposed along the sides of the ice stream, potentially slowing down ice flow. Wide ice streams, on the other hand, are less affected by the no-sliding constraint imposed on slow moving ice, because the constraints are imposed further away. The modeled temperature profile in the AIS/WIS will be different for the IVz and IVz-nosliding groups because it is a narrow ice stream.

"Line 227: The AIS/WIS and BIS boreholes are located in the fast flow region of the Siple coast, where the ice velocities are 365 m yr$^{-1}$ for AIS/WIS-1991-1, 379 m yr$^{-1}$ for AIS/WIS-1995-4,7, and 376 m yr$^{-1}$ for BIS-1998-4,5 from Rignot (2017). The average misfit value of the IVz group is 1.38°C for AIS/WIS-1988-1, 2.16°C for AIS/WIS-1995-4,7, and 0.86°C for BIS-1998-4,5 (Figure 3). In these regions, both IVz and IVz-nosliding allow for basal sliding. However, there are differences in misfit values between IVz and IVz-nosliding groups at AIS/WIS. The reason for these differences is that the modeled ice velocities of IVz-nosliding in the AIS/WIS region are slower than the ones from IVz because it is a narrow ice stream that is influenced by the no-sliding constraint along its sides , resulting in higher misfit values for IVz-nosliding compared to the IVz group."

[Figure]

**Figure 1. The velocity misfit between mean modeled and observed velocity from Rignot (2017). Each magenta dot indicates borehole location, and magenta contour line indicates observed ice velocity with 10 m yr$^{-1}$.**

☑ 233-36: Please explain why the temperatures depend on the quality of the bed geometry.

This paragraph is changed as follows:

[revised manuscript text omitted]

☑ 300 " which of value"—I'm not sure what this phrase indicates.

"which of value" was meant to indicate the original value of total grounded ice melting volume of Llubes et al. (2006) with 16 km yr$^{-1}$. This description seems redundant, therefore, we remove this sentence.

> "Line 302: It is lower than 65 Gt yr$^{-1}$ from Pattyn (2010) and higher than 14.7 Gt yr$^{-1}$ from Llubes et al. (2006), which of value is converted to volume from total ice volume of 16 km3 yr 1 in ice equivalent."

☑ 305: Shouldn't a linear vertical velocity profile be appropriate as long as the surface velocity is approximately equal to the sliding velocity? It seems like this should only result in small errors in the

melt rate

Thank you for raising this point. Yes, if the basal ice velocity is equal to the surface ice velocity in fast-flow regions, the vertical velocity is proportional to the linear vertical profile. However, its basal vertical velocity is constrained as $v_z(b) = v_x(b)\frac{\partial b}{\partial x} + v_x(b)\frac{\partial b}{\partial x} - M_b$ (where $M_b$ represents the basal melting rate).

Furthermore, Joughin et al. (2009) used a thermal model that was not based on an enthalpy formulation, and its method differs from the one used in this study. It is challenging to determine which procedures contribute to the differences between their study and our work. We have removed this sentence from the revised manuscript to not introduce any confusion.

☑ 314: "does not coincide with physical property"—please edit

Below your comment about "Line 310-315, this paragraph requires concision and English grammar." The revised paragraph is listed in below.

☑ 315 "which of equation"—please edit

Below your comment about "Line 310-315, this paragraph requires concision and English grammar." The revised paragraph is listed in below.

☑ 310-325: This paragraph needs rewriting for concision and English grammar.

Upon reviewing the sentence, we realize that it contains redundant information and lacked concision. Taking this into account, we have revised this paragraph as follows.

"Line 308-317: The thermal models have been employed to explore the thermal regime of ice and estimate basal melting rates beneath grounded ice. In the thermal model's advection term, the horizontal components of the ice velocity are estimated using the stress balance equations, whereas the vertical velocity is recovered from the ice incompressibility. Under kriging-based bed topography, the vertical velocity in fast flow regions leads to large flux divergences (Seroussi et al., 2011). In contrast, mass conservation-based bed geometries, such as BedMachine (Morlighem et al., 2017, 2020), preserve low flux divergence. We confirm that the vertical velocity in bed geometry inferred from mass conservation provides a viable way of computing temperature profiles in the Siple coast

fast flow regions, such as the BIS. Additionally, we expect this study to provide a reliable understanding of temperature profiles in the other fast flow regions generated with mass conservation. We should highlight that the good agreement between modeled and observed temperatures in fast flow regions does not guarantee that the magnitude of basal melting volume is accurate, as it depends on both geothermal heat fluxes and frictional heat."

---

## Author Response (AR3)

**Impact of boundary conditions on the modeled thermal regime of the Antarctic ice sheet**
**– Response to Editor's Review –**

We would like to express our gratitute to Dr. Benjamin Smith for his constructive review, which has significantly contributed to the enhancement of our manuscript. We have addressed his comments in a point-by-point manner below. The editor's remarks are indicated in red, and our responses are provided in black (RC).

Public justification (visible to the public if the article is accepted and published):

Line numbers refer to the Authors' Tracked Changes.

199-200: Please check wording. The model results probably suggest something about the glacier, not about the experiments.

Done.

246: consider rewriting as: "plays a more important role in the thermal model than does diffusion"

Done.

247: "The primary difference between BIS..." - it might be better to say "A likely explanation for the difference in the misfits between BIS..."

Done.

306: The phrase as written (which I may have partially provided in my last report on this manuscript) is not entirely correct. I'd suggest rewriting this sentence as something like: "This history results in colder temperatures in the upper part of the ice column, which contains ice that was deposited farther upstream where the surface temperature was lower than it is at the current location of the boreholes. This ice was then transported downstream to the current location." The best citation for

this idea that I've found is a very recent paper:

Hills BH, Christianson K, Jacobel RW, Conway H, Pettersson R (2023).
Radar attenuation demonstrates advective cooling in the Siple Coast ice streams. Journal of Glaciology 69(275), 566–576. https://doi.org/10.1017/jog.2022.86

Done.

339-341: I suggest:
"We confirm that in areas where the bed geometry was inferred from mass continuity, the more accurate estimates of the vertical velocity provide a viable input for estimates of temperature profiles, for example in the Siple Coast fast-flow regions."

Done.

The editing team, Polina Shvedko, noted that the color schemes in Figure 5, representing the geothermal heat flux, basal melting rate, and the difference between basal melting rates, seem similar, impacting readability.

Notification to the authors:
Regarding the figure 5: please ensure that the colour schemes used in your maps and charts allow readers with colour vision deficiencies to correctly interpret your findings. Please check your figures using the Coblis – Color Blindness Simulator (https://www.color-blindness.com/coblis-color-blindness-simulator/) and revise the colour schemes accordingly.

Therefore, Figure 5 has been revised as follows:

[Figure]

Figure 5. Upper panels (a-d) are the geothermal heat flux distributions of each source. Middle panels (e-l) are the basal melting rate distributions, with the value at the bottom left indicating the total grounded ice melting volume for each experiment. The basal melting rate exceeding 50 mm yr⁻¹ is truncated. Lower Panels (m-q) are difference in basal melting rate between IVz-nosliding and IVz for each geothermal heat flux. A green cross dot on the geothermal heat flux map indicates the borehole location. The color maps for the geothermal heat flux and the difference in basal melting rates are from Crameri et al. (2020).